# Type III secretion system effector YfiD inhibits the activation of host poly(ADP-ribose) polymerase-1 to promote bacterial infection

Mengqing Zhou[1], Yabo Liu[1], Yibei Zhang[1,2,3], Yue Ma[1,2,3], Yuanxing Zhang[2,3,4], Sang Ho Choi [5], Shuai Shao [1,2,3✉] & Qiyao Wang [1,2,3,4,6]

Modulation of cell death is a powerful strategy employed by pathogenic bacteria to evade host immune clearance and occupy profitable replication niches during infection. Intracellular pathogens employ the type III secretion system (T3SS) to deliver effectors, which interfere with regulated cell death pathways to evade immune defenses. Here, we reveal that poly(-ADP-ribose) polymerase-1 (PARP1)-dependent cell death restrains *Edwardsiella piscicida*'s proliferation in mouse monocyte macrophages J774A.1, of which PARP1 activation results in the accumulation of poly(ADP-ribose) (PAR) and enhanced inflammatory response. Moreover, *E. piscicida*, an important intracellular pathogen, leverages a T3SS effector YfiD to impair PARP1's activity and inhibit PAR accumulation. Once translocated into the host nucleus, YfiD binds to the ADP-ribosyl transferase (ART) domain of PARP1 to suppress its PARylation ability as the pharmacological inhibitor of PARP1 behaves. Furthermore, the interaction between YfiD and ART mainly relies on the complete unfolding of the helical domain, which releases the inhibitory effect on ART. In addition, YfiD impairs the inflammatory response and cell death in macrophages and promotes in vivo colonization and virulence of *E. piscicida*. Collectively, our results establish the functional mechanism of YfiD as a potential PARP1 inhibitor and provide more insights into host defense against bacterial infection.

[1] State Key Laboratory of Bioreactor Engineering, East China University of Science and Technology, Shanghai, China. [2] Shanghai Engineering Research Center of Maricultured Animal Vaccines, Shanghai, China. [3] Laboratory of Aquatic Animal Diseases of MOA, Shanghai, China. [4] Laboratory for Marine Fisheries Science and Food Production Processes, Qingdao National Laboratory for Marine Science and Technology, Qingdao, China. [5] National Research Laboratory of Molecular Microbiology and Toxicology, Department of Agricultural Biotechnology, Seoul National University, Seoul, Republic of Korea. [6] Shanghai Haosi Marine Biotechnology Co., Ltd, Shanghai, China. ✉email: shaoscott@ecust.edu.cn

Bacterial infection is a complicated antagonism between bacteria and host immune systems, with a consequence of the clearance of bacteria or systemic bacterial infection. Intracellular bacteria, either free in the cytosol or residing in bacteria-containing vacuoles, have evolved a variety of powerful weapons targeting host-regulated cell death pathways to occupy intracellular niches in macrophage, which is believed to be the circulating immune cell accounted for intracellular pathogen proliferation[1].

Upon infection, pathogen-associated molecular patterns, including flagellin, LPS, and DNA/RNA, trigger host defensive elimination mainly depending on regulated cell death-accompanied niche disruption and inflammatory clearance[1]. To escape host immune clearance, bacteria secretion systems deliver sophisticated effectors to disturb regulated cell death in an individual or crosstalk manner. During the infection of *Shigella flexneri*, OspC3, and IpaH7.8 could be delivered through the Type III secretion system (T3SS) to covalently modify caspase-11 and cleave GSDMD, respectively, to evade pyroptosis[2,3]. The T3SS effector SopF of *Salmonella* Typhimurium blocks the V-ATPase-ATG16L1 axis to inhibit xenophagy, promoting in vivo proliferation[4].

At the late stage of infection with the high infective load, however, pathogens residing in macrophages induce regulated cell death to kill infected cells for their pathogenicity and systemic dissemination. The translocated tuberculosis necrotizing toxin of *Mycobacterium tuberculosis* activated receptor-interacting serine-threonine kinases 3 and modulated mixed-lineage kinase domain-like protein to induce host necroptosis, enhancing mycobacterial replication[5]. Simultaneously, the protein tyrosine phosphatase A triggers ferroptosis, a lipid peroxidation-driven regulated cell death, to promote Mtb dissemination[6].

Apart from that apoptosis, necroptosis, and pyroptosis mentioned above are well-known regulated cell death pathways, poly(ADP-ribose) (PAR) polymerase-1 (PARP1) has been characterized to participate in regulated cell death[7,8]. As a nuclear enzyme, PARP1 acts as a DNA damage sensor and attaches up to 200 units of ADP-ribose from $NAD^+$ to a variety of acceptor proteins, including itself[9]. In the presence of DNA damage events, the moderate level of PAR loosens the condensed nucleosome due to its negative charge, rendering DNA accessible to DNA repair enzymes[10,11]. Once encountering excessive DNA damage resulting from acute tissue injuries, virus infection, and bacteria-derived LPS, hyper-activation of PARP1 leads to the 10-500-fold accumulation of nuclear PAR, which is translocated to the cytoplasm serving as a cell death signal and leads to regulated cell death with the membrane rupture[12–14].

Apart from regulated cell death, PARP1 and accumulated PAR have been proposed to participate in the immunologic defense and inflammatory response. The absence of functional PARP1 reduced LPS-induced lung damage in mice, including cytokines expression, alveolar neutrophil accumulation, and lung hyperpermeability[15]. In peritoneal macrophages, PARP1 controls NK cell recruitment to the site of vaccinia viral infection, by promoting the production of chemokine CCL2[16]. As a damage-associated molecular pattern (DAMP), engulfed PAR by macrophages drives inflammatory signaling and stimulates the secretion of TNFα, MCP-1, eotaxin, MIP-1α, and MIP-1β[17]. Moreover, genetic deletion or pharmacological inhibition of PARP1 by the clinically approved PARP inhibitors, Olaparib and Rucaparib, efficiently rescues oncological and inflammatory diseases[18].

*Edwardsiella piscicida* (previously named *E. tarda*), an intracellular bacteria with a wide host range, can survive within macrophages to evade host immune defense and lead to systemic inflammation[19,20]. Here, we identified a T3SS-injected effector YfiD that interacts with PARP1. Given the dual roles of PARP1 as a DNA damage sensor and a mediator of inflammation and regulated cell death, PARP1 participates in the host immune defense against *E. piscicida*'s infection. YfiD rescues the PAR formation by inhibiting the enzymatic domain of PARP1, thereby promoting in vivo colonization and systemic dissemination of *E. piscicida*. This work highlights the critical role of pathogenic effectors in the precise modulation of the regulated cell death pathway and PARylation-accompanied inflammatory response in the host.

## Results

### YfiD is secreted in a T3SS-dependent manner and translocates into host cells.

The implementation of diverse pathogenic strategies mainly depends on T3SS and T6SS to translocate effectors from enteric bacteria to host cells[21]. SDS-PAGE analysis of extracellular proteins (ECPs) of *E. piscicida* pointed out an unidentified protein band below EvpC when cultured in DMEM, the T3/T6SS inducing medium (Fig. 1a, Fig. S3a). Mass-spectrometry-based identification of this protein band indicated several proteins, including the previously identified effectors EvpP, EvpQ, EseG, and TrxlP, implying the presence of novel T3/T6SS-delivered effectors (Fig. S1a)[22–24]. The representative of the identified unique peptides is shown in Fig. S1b. To validate the secretion properties of these proteins identified by mass spectrum, western blot analysis was conducted to detect the presence of these flag-fused proteins in the culture pellet and supernatant of *E. piscicida* WT, ΔT3SS, and ΔT6SS strains. Of them, a 14-kDa protein ETAE_2732 annotated as YfiD (also termed as GrcA, UniProt D0ZD05) was secreted into the culture supernatant depending on T3SS rather than T6SS (Fig. 1b, Fig. S3b). Notably, YfiD augmented in the pellet of ΔT3SS, which might be due to the impaired secretion machine. While YfiD was accumulated in the secretome of ΔT6SS and this feature probably was the consequence of T3SS-T6SS interaction because the absence of T6SS might save energy to promote protein secretion via T3SS in bacteria.

We further examined whether YfiD could be translocated into host cells via TEM-1 β-lactamase reporter assay (Fig. S1c)[4]. Initially, a plasmid containing the YfiD-TEM-1 fusion was engineered and transformed into *E. piscicida* WT, ΔT3SS, and ΔT6SS strains, respectively, to verify the translocation mode of YfiD. The intracellular translocation of YfiD was observed in approximately 16% or 17% of HeLa cells infected with wild-type *E. piscicida* or T6SS mutant, respectively, while the positive signals (blue) nearly disappeared (approximately 1.3%) in those infected with T3SS mutant. These secretion profiles of YfiD were consistent with those of T3SS effector EseG (Fig. 1c and d)[25]. Moreover, multiple alignments of YfiD in several gram-negative pathogens including *Klebsiella*, *Salmonella*, *Shigella*, and *Edwardsiella* revealed the consistency in amino acids composition, especially the carboxyl-terminal (Fig. S1d). The high similarity between YfiD and the Gly_radical domain of pyruvate formate-lyase PflB in *E. piscicida* was determined, which cleaves pyruvate to produce formate and acetyl-CoA (Fig. S1e)[26]. We thus hypothesized that YfiD potentially influences the key metabolic pathways linked to metabolic disease in the host. Taken together, these results demonstrated that YfiD conserved in gram-negative pathogens is a novel T3SS effector and is translocated into host cells in a T3SS-dependent manner.

### YfiD interacts with the ART domain of PARP1 to suppress PAR formation.

To figure out the intracellular function of YfiD, lentivirus-mediated stable expression of HA-tagged YfiD was generated in the HeLa cell line, which was validated by western blot analysis (Figs. S2a and S3h). Subsequently, the lysates of

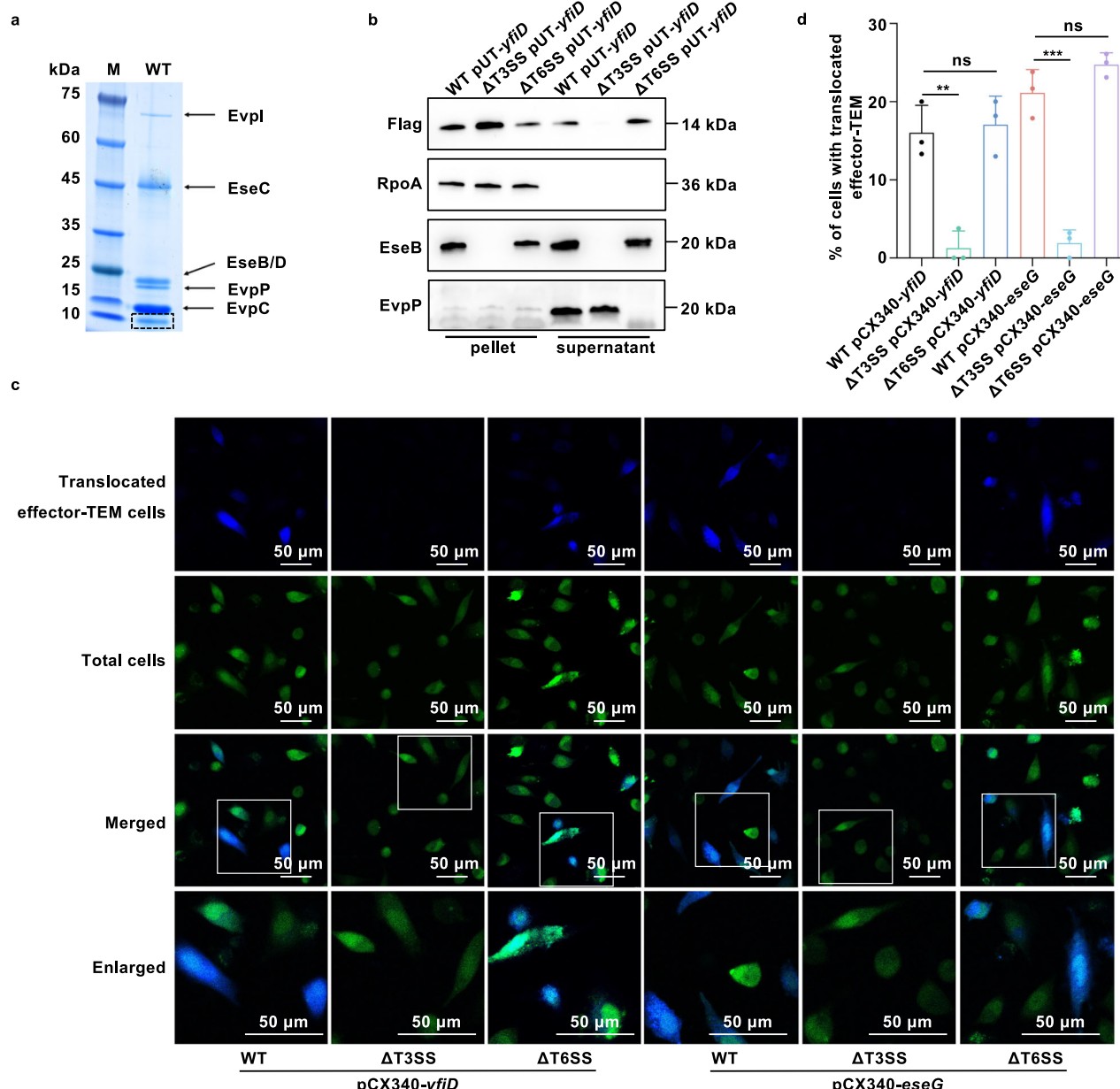

**Fig. 1 Characterization of YfiD as a T3SS effector. a** Protein secretion profile of *E. piscicida* WT grown in DMEM at 30 °C. The appearance of one band at approximately 10 kDa indicates the potential extracellular proteins (ECPs). The supernatant was concentrated by ultrafiltration and analyzed by SDS-PAGE with coomassie blue staining. **b** Secretion profile of YfiD in the T3SS-dependent manner. The presence of YfiD in ECPs and bacteria pellets of WT, ΔT3SS, and ΔT6SS was detected by western blot analysis. YfiD was probed with anti-Flag antibody, while RpoA, EseB, and EvpP were served as loading controls. **c–d** Translocation of YfiD into host cells. HeLa cells were infected with WT, ΔT3SS, or ΔT6SS carrying a pCX340-*yfiD*-TEM or pCX340-*eseG*-TEM plasmid at MOI 100 and incubated with CCF2-AM at 6 hpi. Images were captured by confocal microscopy and processed by Image J (**c**). The percentage with translocated effector-TEM (blue) cells was determined by counting the ratio of blue cells to green cells in the foci (**d**). EseG, a T3SS effector, was used as a positive control. All images shown are representative of at least three independent experiments. The results were shown as the mean ± S.D. (*n* = 3). ***, *P* < 0.001; **, *P* < 0.01; ns, non-significance, *P* > 0.05 based on two-tailed Student's *t*-test.

HeLa-YfiD cells were immunoprecipitated with anti-HA beads followed by mass spectrometry analysis. Compared with the immunoprecipitates in HeLa-Mock cells, a scatter plot was drawn according to the number of unique peptides in which the ratio was calculated as spectral counts to reflect preferential detection in HeLa cells expressing YfiD (Fig. 2a). Of these, the ranked ten proteins were listed and most of the ten candidates were associated with cytoskeletal rearrangement (Fig. 2b). Notably, PARP1 was reported to participate in host-pathogen interactions, inflammation, regulated cell death, and tumorigenesis[27]. The

representative of the identified unique peptides of PARP1 has been included in Figure S2b.

As a multi-domain DNA damage sensor, PARP1 (UniProt Q921K2) consists of three zinc-finger motifs (F1, F2, and F3) recognizing DNA breaks and nicks, an auto modification domain (BRCT) targeted by self-generated PAR, a Trp-Gly-Arg domain (WGR) indispensable for allosteric activation, and a catalytic domain (CAT) in which the helical domain (HD) functions as an auto-inhibitory region for ADP-ribosyl transferase (ART) domain. In the absence of DNA damage, HD occupies the

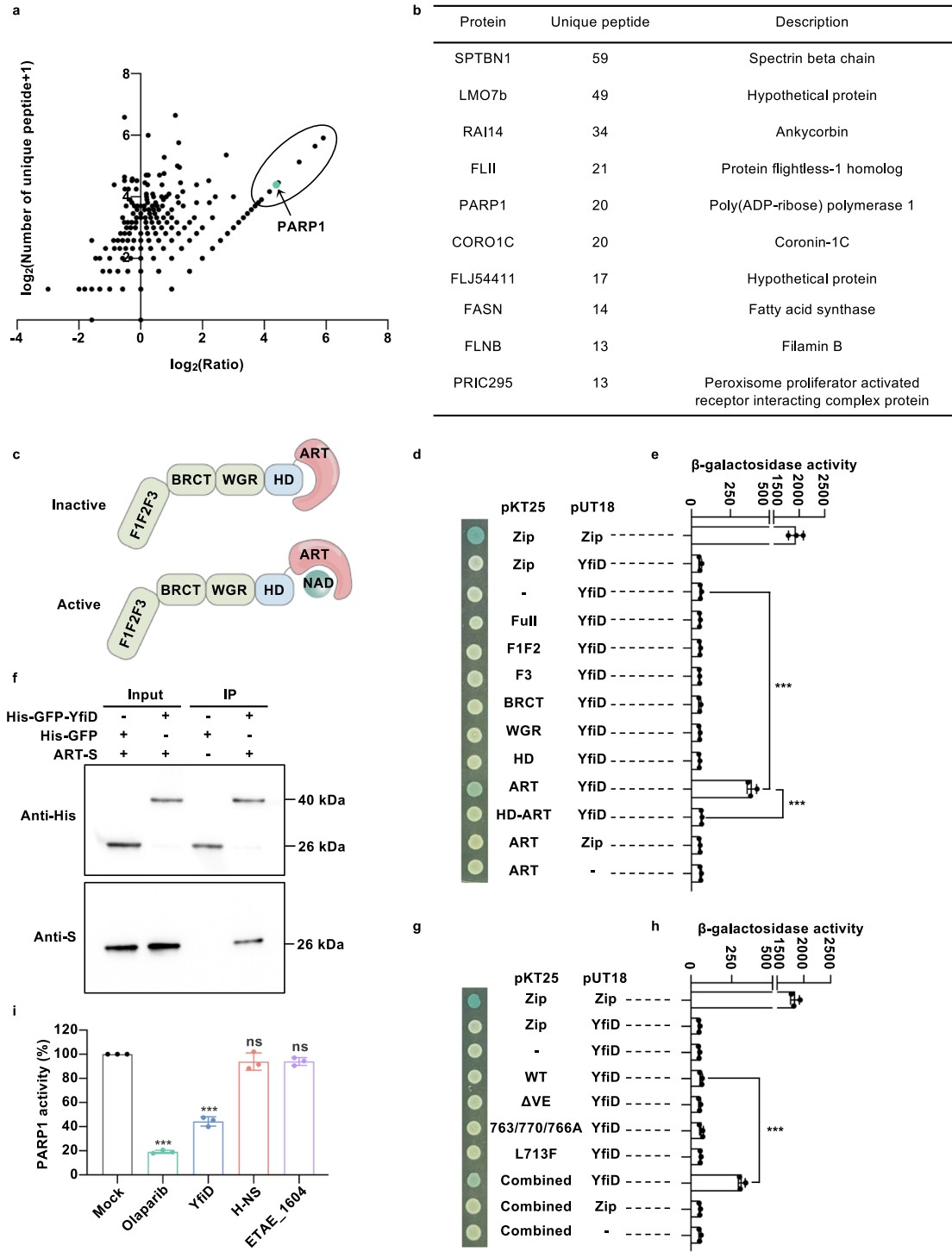

NAD$^+$-binding site of ART domain to suppress the enzymatic activity of PARP1. Along with the occurrence of DNA damage, unfolded HD releases from the ART domain to allow NAD$^+$ approaching and PARP1 activation (Fig. 2c)[28,29]. To investigate the interaction between YfiD and PARP1, the bacterial adenylate cyclase two-hybrid (BACTH) system was performed, which is a well-known approach to characterize protein-protein interactions in vivo[30]. Using the leucine zipper (Zip) of transcription activator GCN4 as the positive control, the co-transformants of F1F2, F3, BRCT, WGR, and HD with YfiD did not produce any detectable signal (white), respectively. In the presence of X-Gal, the interaction between ART and YfiD exhibited a strong positive signal (blue) (Fig. 2d). Consistently, using 2-Nitrophenyl β-D-galactopyranoside (ONPG) as a colorimetric substrate, ART-YfiD generated a detectable interaction in a β-galactosidase assay as well, while F1F2, F3, BRCT, WGR, and HD did not (Fig. 2e). However, the failed detection of the interaction between the full-length PARP1 and YfiD probably due to the presence of HD. The disappeared interaction between HD-ART and YfiD further confirmed that HD might be embedded in ART and disrupt the interface between ART and YfiD (Fig. 2d-e). In addition, a GFP beads-mediated pull-down assay was conducted to support the interaction between ART and YfiD that S-tagged ART specifically bounds to GFP-YfiD, but not GFP (Fig. 2f, Fig. S3c).

**Fig. 2 YfiD interacts with PARP1 to suppress its enzymatic activity. a** The scatter plot of protein ratios as a function of their relative abundance. HeLa cells were stably transfected with pCDH-YfiD-HA or pCDH, respectively. Cell lysates were subjected to anti-HA immunoprecipitation and analyzed by mass spectrometry. The ratio was calculated as the number of unique peptides in HeLa-YfiD divided by those in HeLa-Mock immunoprecipitates. Dots in the circle correspond to the top ten proteins specific in HeLa-YfiD, while the green dot corresponds to PARP1. **b** The top ten YfiD-interacting proteins are listed according to the number of unique peptides. (**c**) Schematic representation of the inactivated and activated PARP1, consisting of three zinc-finger motifs (F1, F2, and F3), an auto modification domain (BRCT), a Trp-Gly-Arg domain (WGR), and a catalytic domain (CAT) containing the helical domain (HD) and ADP-ribosyl transferase (ART) domain. **d–e** Determination of bacterial two-hybrid interactions between YfiD and full-length and truncated PARP1. The β-galactosidase activity was observed based on X-gal/IPTG (d) and further measured with ONPG-dependent colorimetric analysis (**e**). Zip-Zip interaction was served as a positive control, while Zip-YfiD and ART-Zip were used as negative controls. **f** Interaction between YfiD and ART by GFP beads-mediated pull-down assay. Recombinant protein His-GFP or His-GFP-YfiD was mixed with S-tagged ART and then immobilized to anti-GFP magnetic beads. After eluted with glycine-elution buffer, the samples were analyzed with anti-His and anti-S antibodies. **g–h** Determination of bacterial two-hybrid interactions between YfiD and HD-ART variants. The β-galactosidase activity was observed based on X-gal/IPTG (g) and further measured with ONPG-dependent colorimetric analysis (h). Zip-Zip interaction was served as a positive control, while Zip-YfiD and Combined-Zip were used as negative controls. **i** In vitro PARP1 chemiluminescent assay. The enzymatic activity of purified PARP1 was quantified with histone substrates, activated DNA, and tested inhibitors. Olaparib, purified HNS, and T3SS effector ETAE_1604 were served as controls, respectively. The image shown is representative of at least three independent experiments. The results were shown as the mean ± S.D. ($n = 3$). ***, $P < 0.001$; ns, non-significance, $P > 0.05$ based on two-tailed Student's $t$-test.

During the activation process of PARP1, the HD undergoes an unfolding process to release the inhibitory effect on the catalytic domain, which mainly occurs in the α-helixes contacting the ART domain (Fig. S2c)[29]. The deletion of Val687/Glu688 (ΔVE) in αB and the substitutions of L713 (αD) and E763/D766/D770 (αF) led to the conformational destabilization of HD, which thus enhanced DNA-independent PARP1 catalytic activity[31,32]. To further explore the HD's blockade of ART-YfiD, HD-ART polypeptide variants were expressed in fusion with T25 fragment for BACTH assay, which contained the individual or combined mutations as mentioned above (Fig. S2c). In the presence of X-gal or ONPG for β-galactosidase assay, the separated co-transformants of ΔVE, L713F, or E763/D766/D770A, respectively, with YfiD did not produce any detectable signal, while the combined mutations interacted with YfiD to generate a positive signal (Fig. 2g and h). Based on the observations, we supposed that YfiD contacts with the ART domain of PARP1 in the presence of the complete unfolding of HD.

To investigate the outcome of the interaction between YfiD and the catalytic domain of PARP1, the PARP1 chemiluminescent assay was employed with purified PARP1 enzyme, histone substrates, and activated DNA to quantity PARP1 activity in vitro[33]. The presence of Olaparib and purified YfiD declined PARP1 activity to approximately 19% and 45% compared to that without any treatment, respectively (Fig. 2i). As a non-related control, H-NS and another T3SS effector had no effect on PARP1 activity. These data confirmed that YfiD inhibits PARP1 activity induced by DNA lesions. Taken together, YfiD interacts with the catalytic domain of PARP1 to inhibit PARP1 activation and PAR formation.

### *E. piscicida*-delivered or stable-expressed YfiD inhibits PAR formation in macrophages.

Given that infection of many pathogens, including ZIKV virus[13] and *Streptococcus pyogenes*[34] induces PARP1 activation and PAR formation in host cells, we initially explored whether the infection of *E. piscicida* results in the accumulation of intracellular PAR mainly synthesized by PARP1 in macrophage J774A.1 cells. The anti-PAR antibody was used along with phosphorylated H2AX (pH2AX) indicating DNA damage events[35]. Before the treatment, PAR signals were almost undetectable, which is indicative of the inactivated PARP1. The formation of PAR was observed in *E. piscicida*-infected J774A.1 and reached a peak with the occurrence of DNA damage at 2 hours post-infection (Fig. 3a, Fig. S3d). Notably, the accumulated effect of PAR was alleviated by the PARP inhibitor Olaparib rather than the pan-caspase inhibitor Z-VAD-FMK (Fig. 3a,

Fig. S3d). Although PARP1 is a substrate of activated caspases in caspase-dependent apoptosis[18], PAR formation was not influenced by caspase activation during *E. piscicida*'s infection.

Given that YfiD impairs PARP1 enzymatic activity in vitro, whether YfiD impairs PAR formation in vivo remains intriguing. The *yfiD* deletion mutant was constructed based on homologous recombination to investigate the effect of YfiD on PAR formation during infection. The presence of PAR was determined upon infection of wild-type (WT) *E. piscicida* or *yfiD* deletion mutant Δ*yfiD* (Fig. 3b, Fig. S3e). Either cultivated in LB medium or static opti-MEM mimicking in vivo condition, no growth defect was observed in Δ*yfiD* (Fig. S2d and e). In the absence of YfiD, the accumulated PAR caused by bacterial infection was approximately 1.5-fold (1.9 vs 1.3 of relative density) and 1.6-fold (3.8 vs 2.4 of relative density) more than those infected with WT during the early 1 or 2-hour post-infection, respectively, suggesting that YfiD suppresses PAR formation during infection. Afterward, the level of PAR infected with Δ*yfiD* was recovered to that infected with WT at 3 hpi. Meanwhile, pH2AX-marked DNA damage events slightly augmented in Δ*yfiD*-infected cells compared with WT-infected cells.

To further confirm the notion that YfiD impacts PAR accumulation, a J774A.1 cell line was engineered to express YfiD under the control of a CMV promoter (Fig. S2f and S3i) and was exposed to N-methyl-N'-nitro-N-nitrosoguanidine (MNNG). As a well-defined DNA-alkylating agent, MNNG has been widely used to activate PARP1[14]. The exposure to gradient concentration of MNNG for 15 min induced the generation of a great amount of PAR and pH2AX in the mock group, whereas the presence of YfiD conferred J774A.1 resistance to MNNG at a certain level. When 200 μM MNNG was used, for example, the relative density of PAR declined from 9.6 in the mock group to 4.2 in the group continuously expressing YfiD. Intriguingly, the relative density of pH2AX declined from 4.1 in the mock group to 3.0 in the group continuously expressing YfiD (Fig. 3c, Fig. S3f). Congruent with *E. piscicida*'s infection, these results further supported that YfiD suppresses PAR formation caused by DNA alkylation.

### YfiD attenuates PARP1-dependent cell death.

Excessive stimulation of PARP1 and PAR accumulation are considered the key features of PARP1-dependent cell death and its remarkable morphology is the rupture of plasma membranes[8]. Considering the inhibitory effect of PARP1 activity and PAR formation by YfiD, we hypothesized that YfiD can alleviate PARP1-dependent cell death in macrophages. To this end, a propidium iodide (PI) uptake assay was utilized to evaluate the influence of YfiD on host

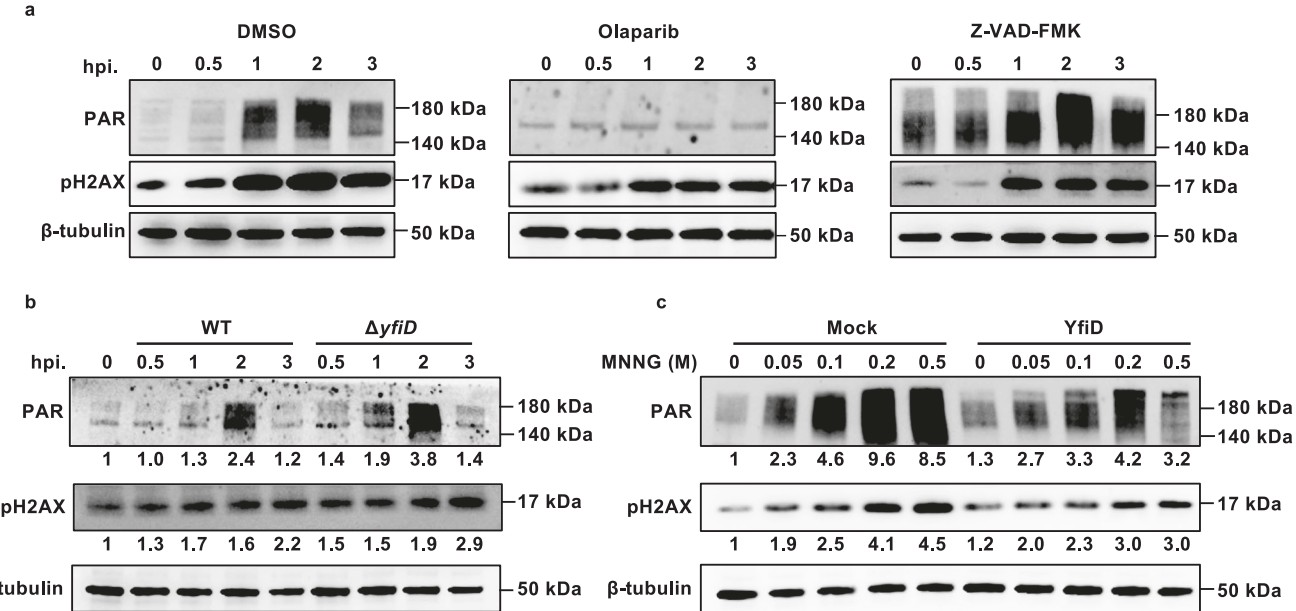

**Fig. 3 YfiD inhibits PAR formation in macrophages. a** Accumulation of PAR during *E. piscicida*'s infection. Following the pretreatment of DMSO, Z-VAD-FMK, and Olaparib, J774A.1 cells were infected with WT. At each indicated time, phosphorylation of H2AX and PAR accumulation were analyzed by Western blot assay. β-tubulin was used as a loading control. **b** YfiD suppresses PAR formation during infection. J774A.1 cells were infected with *E. piscicida* WT or Δ*yfiD*, respectively. At each indicated time, PAR accumulation and phosphorylation of H2AX were analyzed by Western blot assay. β-tubulin was used as a loading control. **c** YfiD suppresses PAR formation in MNNG-treated J774A.1 cells. After the treatment of 0, 50, 100, 200, or 500 μM MNNG for 15 min, PAR accumulation and phosphorylation of H2AX in J774A.1 cells expressing YfiD and the mock group were determined by Western blot assay. β-tubulin was served as a loading control. The image shown is representative of three independent experiments (*n* = 3).

cell membrane integrity. The PARP1 inhibitor Olaparib, and the pan-caspase inhibitor Z-VAD-FMK, were used as positive and negative controls, respectively. In the presence of 200 μM MNNG, the J774A.1 expressing YfiD or not exhibited impaired cell membrane integrity in a time-series manner in comparison to DMSO-treated J774A.1, indicating that MNNG induced cell death with the membrane rupture in J774A.1 cells. The treatment of Olaparib significantly rescued cell death caused by MNNG, while Z-VAD-FMK had no impact on cell viability. These observations suggest that MNNG-induced DNA lesion triggered PARP1-dependent cell death in mouse monocyte macrophages J774A.1. More importantly, stable-expressing YfiD significantly increased the cell viability of MNNG-treated cells compared to the mock group. Besides, the gap in cell viability of the mock and YfiD expression group under the treatment of Olaparib was narrowed compared to the treatment of MNNG (Fig. 4a and b). Taken together, macrophages expressing YfiD are resistant to PARP1-dependent cell death.

Translocation of mitochondrial AIF and cytoplasmic MIF to the nucleus is a well-defined signature in canonical parthanatos[8,36], however, different cell types may employ the AIF-dependent or independent parthanatos. We thus investigated the distribution of AIF and MIF in cell fractions following MNNG treatment. Cells were treated with or without MNNG (200 μM) for 6 h or 8 h, respectively, and then separated into cytoplasmic fractions and nuclear fractions. The location of AIF and MIF was determined by immunoblot, while β-tubulin and Lamin B1 were served as markers for cytoplasm and nucleus, respectively. Regardless of MNNG stimulation, AIF and MIF exhibited apparent localization in cytoplasmic fractions, suggesting that translocation of AIF and MIF is dispensable for PARP1-dependent cell death in J774.1 (Fig. 4c, Fig. S3g). To verify where the interaction between PARP1 and YfiD occurs, the locations of PARP1 and YfiD were further determined as described above. Regardless of MNNG stimulation, PARP1 constantly localized in

nuclear fractions. Notably, YfiD remained in the cytoplasm without stimulation, whereas under the treatment of MNNG, it appeared to robustly translocate to the nucleus and colocalized with PARP1 in a time-dependent manner combined with a boom of expression (Fig. 4c, Fig. S3g). Specifically, YfiD was transported into the nucleus either with a 15-min MNNG treatment or at 2 hours post-Δ*yfiD*-infection (Fig. S2g and S3j). The occurrence of YfiD in the nucleus correlated with the PAR accumulation during the early infection stage of *E. piscicida*. Collectively, these results revealed that cytosolic YfiD translocates to the nucleus to interfere cell death of macrophages during bacterial infection.

**YfiD facilitates the pathogenicity of *E. piscicida*.** Apart from the critical role involved in DNA damage repair and regulated cell death, PARP1 exhibits its proinflammatory effects in multiple dimensions. PARP1 sustains the function of a common set of inflammatory mediators, embracing interleukin-1 (IL-1), interleukin-6 (IL-6), tumor necrosis factor-α (TNF-α), and high mobility group box 1 protein (HMGB1)[17,37]. PAR, the enzymatic product of PARP1, also acts as a DAMP to aggravate the inflammation of macrophages[17]. To determine if *E. piscicida*-stimulated PARP1 activation serves the inflammation-mediated elimination of intracellular bacteria, we examined the transcripts of the above inflammatory mediators in macrophages. Prior to being infected by WT *E. piscicida* or Δ*yfiD*, cells were incubated with or without Olaparib for 2 h and the expression of these inflammatory mediators was determined at 4 hpi. Compared with WT infection, deletion of *yfiD* led to the superior transcription of *IL-1α* (NP_034684.2), *IL-1β* (NP_032387), and *IL-6* (NP_112445.1) (Fig. 5a). Olaparib attenuated the expressions of *IL-1α*, *IL-1β*, and *IL-6* stimulated by Δ*yfiD* which were similar as WT did, implying that YfiD inhibited the proinflammatory response which was promoted with PARP1 activation by *E. piscicida* infection (Fig. 5a).

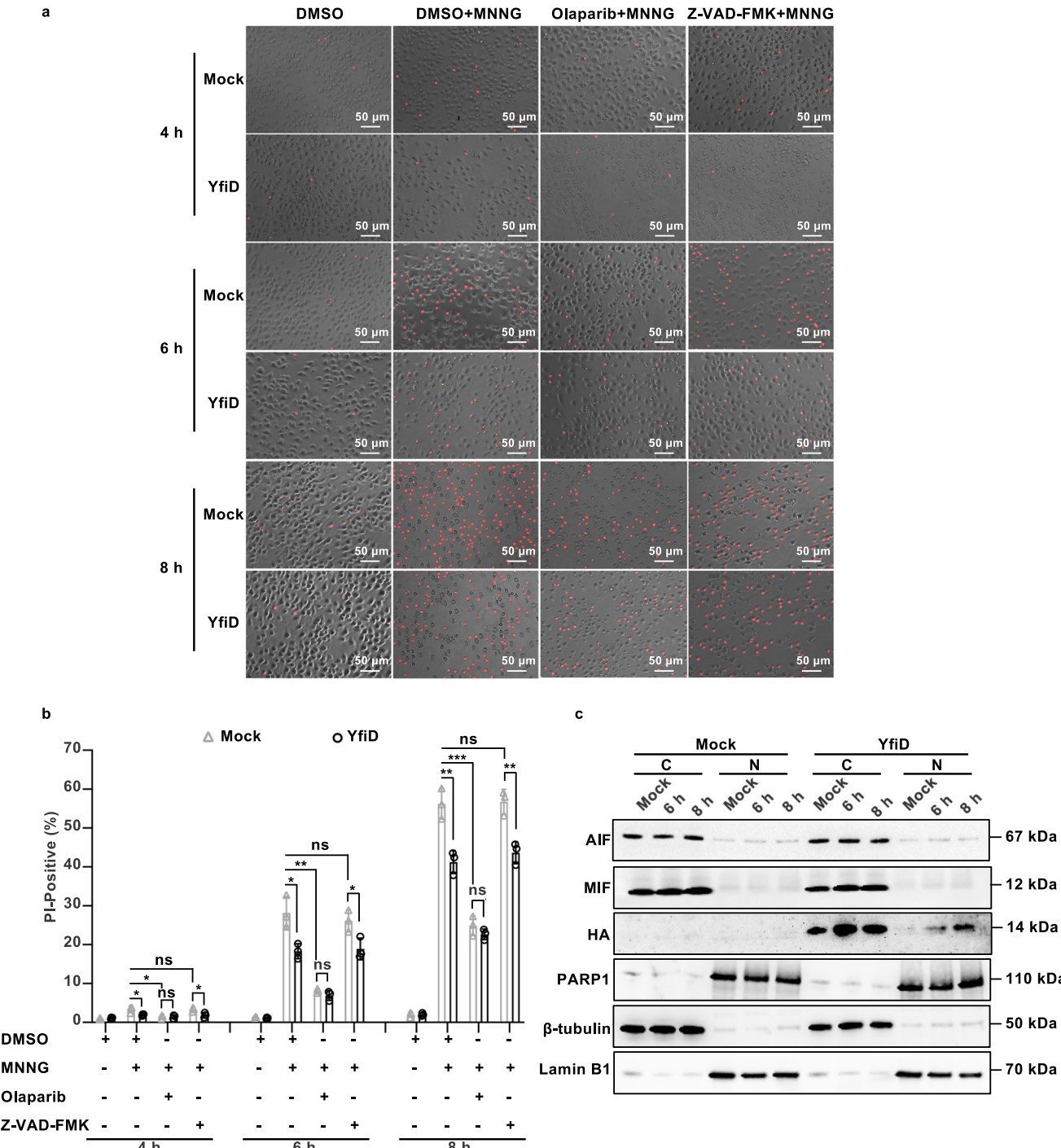

**Fig. 4 YfiD attenuates PARP1-dependent cell death. a, b** YfiD alleviates PARP1-dependent cell death. J774A.1 cells stably expressing YfiD or Mock were pretreated with DMSO, Z-VAD-FMK, and Olaparib prior to MNNG (200 μM) treatment for 15 min. At each indicated time (4, 6, and 8 h), cell viability was analyzed by PI uptake assay (**a**). **b** Statistical analysis of cell viability was determined by counting the ratio of red cells to total cells. The results were shown as the mean ± S.D. ($n = 3$). ***, $P < 0.001$; **, $P < 0.01$; *, $P < 0.05$; ns, non-significance, $P > 0.05$ based on two-tailed Student's $t$-test. **c** YfiD-mediated cell death is independent of AIF and relies on the translocation of YfiD to the nucleus. J774A.1 cells expressing HA-tagged YfiD or Mock were exposed to 200 μM MNNG for 15 min. At each indicated time, the cytoplasmic and nuclear fractions were collected and analyzed with Western blot assay. Subcellular distributions of AIF, MIF, PARP1, and YfiD-HA were examined by the corresponding antibodies, while β-tubulin and Lamin B1 were served as cytoplasmic and nuclear markers, respectively. All images shown are representative of three independent experiments ($n = 3$).

Given that YfiD suppressed PAR accumulation and the expression of proinflammatory cytokines, we speculated PAR accumulation is a strategy employed by macrophages to defend against invasive pathogens and further explored whether the inhibition of PAR formation by YfiD contributes to the pathogenesis of *E. piscicida*. Initially, the cell viability of J774A.1 infected by WT or Δ*yfiD* was determined by PI uptake assay. The absence of YfiD led to apparently more dead cells than WT did, while the treatment of Olaparib eliminated the influence of YfiD on cell viability (Fig. 5b and S2h). Moreover, cells infected

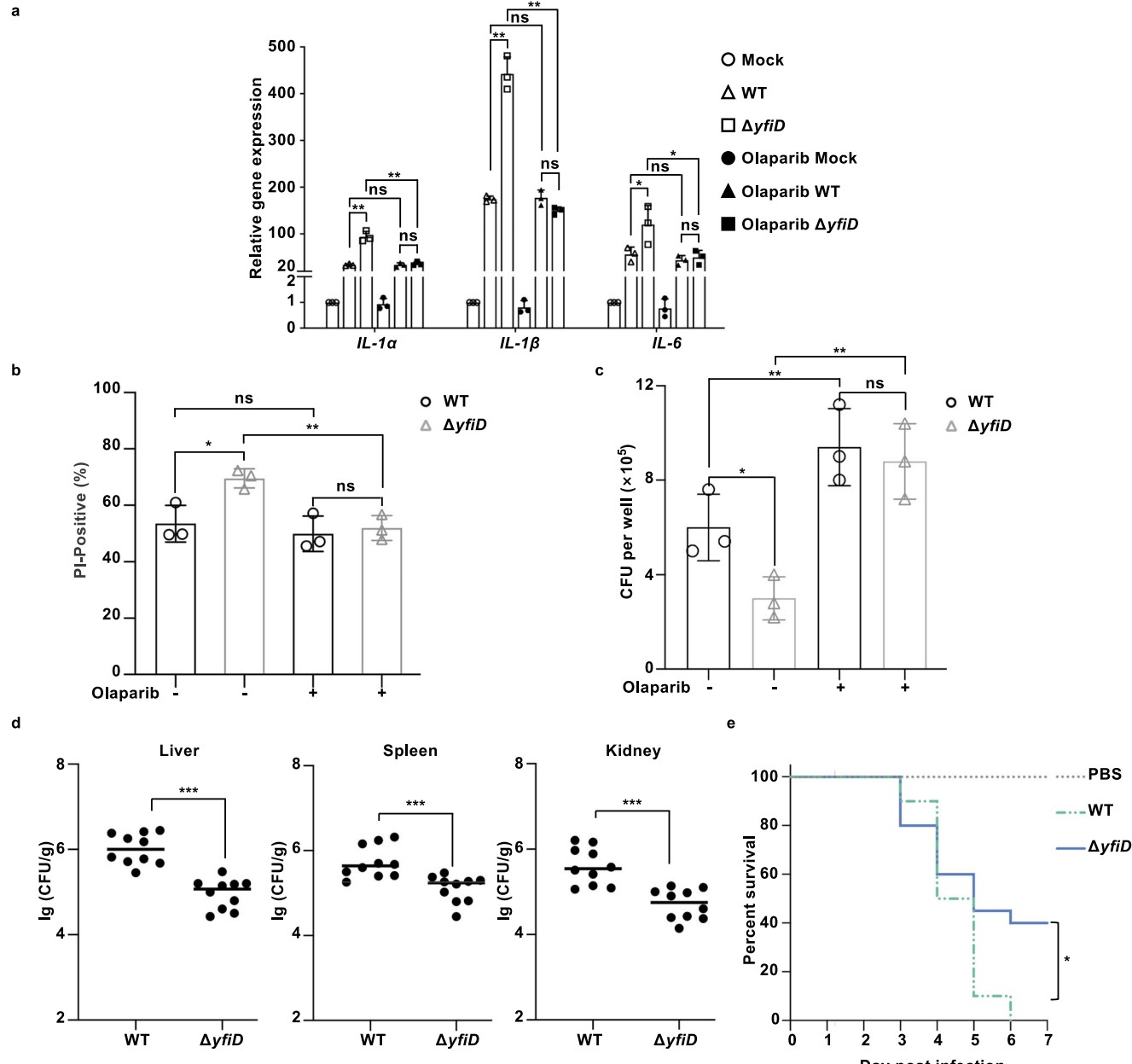

**Fig. 5 YfiD facilitates the pathogenicity of *E. piscicida*. a** YfiD inhibits the expression of proinflammatory genes during infection. Following DMSO and Olaparib pretreatment, J774A.1 cells were infected with *E. piscicida* WT or Δ*yfiD*, respectively. At 4 hpi, transcripts of *IL-1α*, *IL-1β*, and *IL-6* were determined by RT-qPCR. **b** Statistical analysis of cell viability of J774A.1 cells infected with *E. piscicida* WT or Δ*yfiD* in the presence or absence of Olaparib at 6 hpi. **c** Intracellular survival of WT and Δ*yfiD* in J774A.1. Following the pretreatment of DMSO and Olaparib, J774A.1 cells were infected with WT and Δ*yfiD* at MOI 10. The number of intracellular bacteria was determined at 6 hpi. **d** Colonization of WT and Δ*yfiD* in turbots. The infection model was established by intraperitoneal injection of *E. piscicida* WT and Δ*yfiD* in turbots (10 per group) at a dose of 1×10⁶ CFUs/fish. At day 8 post-infection, CFUs from grinded livers, spleens, and kidneys were determined by plating on agar plates. **e** Percent survival of zebrafish (20 per group) was recorded for *E. piscicida* WT and Δ*yfiD* injection at a dose of 400 CFU/fish during a 7-day observation. All images shown are representative of three independent experiments (*n* = 3). ***, *P* < 0.001; **, *P* < 0.01; *, *P* < 0.05; ns, non-significance, *P* > 0.05 based on two-tailed Student's *t*-test and Kaplan-Meier survival analysis with a log-rank test (Mantel-Cox).

by WT exhibited similar cell viabilities regardless of the treatment of Olaparib, suggesting that YfiD functions to inhibit PARP1-dependent cell death as Olaparib does. Furthermore, the intracellular survival capacities of WT or Δ*yfiD* were discriminated, indicating the lower survival rate of Δ*yfiD* than WT in macrophages (Fig. 5c). The treatment of Olaparib restored the intracellular survival of Δ*yfiD* to that of WT, implying that YfiD suppresses PARP1-dependent cell death to facilitate the intracellular survival of *E. piscicida* in macrophages.

To assess the function of YfiD for systemic infection, an intraperitoneal or intramuscular injection model was employed in turbots and zebrafish, respectively. At 8 days post-infection, Δ*yfiD* possessed a lower burden than WT did in livers, spleens, and kidneys of infected turbots (Fig. 5d). Consistently, WT-infected zebrafish displayed higher mortality relative to the Δ*yfiD*-infected zebrafish, indicating the important role of YfiD in promoting bacterial pathogenicity (Fig. 5e). Collectively, lack of YfiD leads to severe regulated cell death and proinflammatory response in

macrophages, which thus disrupts the intracellular niches for the replication of *E. piscicida* and thus disturbs *E. piscicida* dissemination in the host.

## Discussion

Following infection in macrophages, intracellular pathogens, including *E. piscicida* modulate the properties of host cells to occupy niches for colonization and evade host immune clearance[38,39]. In this study, we identified a T3SS effector, named YfiD, to rescue PAR accumulation and PARP1-dependent cell death driven by *E. piscicida*'s infection. Once translocated into the host nucleus, YfiD interacts with the catalytic domain of PARP1 to suppress PAR formation, thus promoting in vivo colonization and virulence of *E. piscicida*.

A few studies have established the connection between PARP1-dependent cell death and infection. Zika virus infection induces canonical PARP1-dependent cell death in HeLa cells, ablating host restriction of virus dissemination[13]. Bacteria-derived LPS was demonstrated to induce canonical PARP1-dependent cell death in BMDM cells through TLR4 receptors and non-canonical PARP1-dependent cell death in CD4 T cells[40,41]. These inspiring studies hint at the role of PARP1-mediated defense against bacterial infection. Herein, we established bacterial infection-induced PARP1-dependent cell death in an AIF-independent manner in mouse monocyte macrophages J774A.1 (Fig. 3 and Fig. 4). Cell death-impelled niche disruption and inflammation restrict the colonization of *E. piscicida* in macrophages (Fig. 5a-c), which is crucial for the host to resist invasive pathogens.

For intracellular pathogens, a key mechanism determining the outcome of infection is the bacterial ability to alter regulated cell death pathways[1]. A T3SS effector YfiD was characterized (Fig. 1), which is conserved among a large number of Gram-negative bacteria as a spare part of PflB (Fig. S1d and e)[26]. YfiD evolutionarily exhibits the prevention of macrophages from PAR accumulation and regulated cell death to generate an available niche (Figs. 3 and 4). Furthermore, the effective suppression of PAR formation relies on YfiD's mode to translocate to the nucleus, in response to DNA damage, where YfiD acts as an inhibitor of PARP1 activation by attaching to its ART domain. While, HD, the autoinhibitory domain of the PARP1, renders YfiD attachment to the ART domain (Figs. 2 and 4c).

At the earlier phase of intracellular survival within macrophages, YfiD inhibits PARP1-dependent and time-series PAR excessive formation, inflammatory response, and regulated cell death, offering a favorable environment for bacterial colonization (Fig. 3 and Fig. 5a-c). At the late phase of intracellular survival within macrophages, however, *E. piscicida* has to break out the limited intracellular space and spread to other cells and tissues[42]. Due to the lack of sufficient intracellular replication, Δ*yfiD* failed to systematically infect the host with an alleviated bacterial burden and lethality to fish (Fig. 5d and e). Therefore, a model of YfiD-PARP1 interaction was proposed (Fig. 6). Following infection of *E. piscicida* lacking *yfiD* in macrophages, internalized bacteria-caused DNA lesion is recognized by PARP1 followed by its self-activation. As an alarming indicator, the activation of PARP1 leads to regulated cell death and inflammatory response, restraining the proliferation of *E. piscicida* in macrophages. Along with the occurrence of pathological stimulation in macrophages, cytosolic YfiD enters into the nucleus, where it inhibits PARP1 activation and PAR formation by interacting with the ART domain of PARP1. This blockade rescues *E. piscicida* from niche disruption and inflammatory response, facilitating *E. piscicida* proliferation in macrophages and further the diffusive infection of *E. piscicida* in the host.

The use of PARP1 inhibitors in cancer therapy is believed to have a prospective future, however, there are few reports about PARP1 inhibitors of biological origin. In this work, we demonstrated the role of YfiD in retarding PAR formation by interacting with the ART domain of PARP1 once translocated to the nucleus. This finding filled a research gap in the connection between bacteria secretion system-injected effectors and PARP1 inhibition. Moreover, YfiD-ART interaction only occurs when the HD of PARP1 undergoes an unfolding process, although the underlying mechanism of YfiD preventing PAR accumulation remains obscure. Further elucidation of YfiD-mediated PARP1 hypoactivity in broad cell types might provide further insights into the structural basis of novel and biological PARP1 inhibitors. In particular, the identification of YfiD opens up new opportunities for the precise development of PARP1 inhibitors that respond to DNA damage-related signals.

## Materials and methods

**Cell culture and treatment**. HEK293T, HeLa, and J774A.1 cells were routinely cultured in Dulbecco's modified Eagle medium (DMEM, Thermo Fisher) supplemented with 10% heat-inactivated fetal bovine serum (FBS, biosera), 1% penicillin-streptomycin (Beyotime) and 3.7 g/L sodium bicarbonate (NaHCO$_3$, Merck) in a humidified 37°C incubator containing 5% CO$_2$. Unless stated otherwise, cells were exposed to 200 μM N-methyl-N'-nitro-N-nitrosoguanidine (MNNG, Abmole) in darkness for 15 min at 37°C before washing three times with PBS. When necessary, cells were incubated with 20 μM Z-VAD-FMK (Beyotime) or 5 μM Olaparib (Selleck) for 2 h prior to MNNG treatment or infection.

**Bacterial strains, plasmids, and culture conditions**. Bacterial strains and plasmids used in this study are listed in Table S1, and the primers used in this study are shown in Table S2. *E. piscicida* strains were grown in Luria-Bertani (LB) broth medium at 37°C or DMEM at 30°C, mimicking in vivo environment. *E. coli* strains were cultured in LB medium at 37°C. When necessary, the following antibiotics were used at the appropriate concentrations: polymyxin (Col, 100 μg/mL), carbenicillin (Carb, 100 μg/mL), chloramphenicol (Cm, 25 μg/mL), and kanamycin (Km, 100 μg/mL).

**Construction of in-frame deletion mutants and complemented strains**. The construction of in-frame knock-out mutants was based on homologous recombination[43]. Briefly, the upstream and downstream sequences were cloned into pDM4 digested by *Xba*I. The recombinant vectors were introduced into *E. coli* SM10 and then conjugated with *E. piscicida*. Double crossover recombinant events were selected on LB plates containing 15% sucrose and then verified by sequencing.

For the construction of *E. piscicida* complemented strains, the fragment containing *yfiD-flag* or *yfiD*-TEM-1 fusion was cloned into plasmid pUT or pCX340, respectively. Then the generated recombinant plasmids were transformed into the corresponding *E. piscicida* strains by electroporation at 2.5 KV for 2.5 ms.

**Bacterial infection, intracellular proliferation, and TEM-1 β-lactamase reporter assay**. HeLa and J774A.1 cells seeded in reduced-serum opti-MEM (Gibco) at a density of 8×10$^4$ cells/cm2 or 1.2×10$^5$ cells/cm2, were infected with the logarithmic phase of *E. piscicida* at an MOI (multiplicity of infection) of 100 or 10, respectively. The concentration of 1000 μg/mL gentamicin (Gm) was added at 2 hpi for 30 min to remove uninternalized bacteria, and then cells were cultured in opti-MEM containing 10 μg/mL Gm for the indicated time.

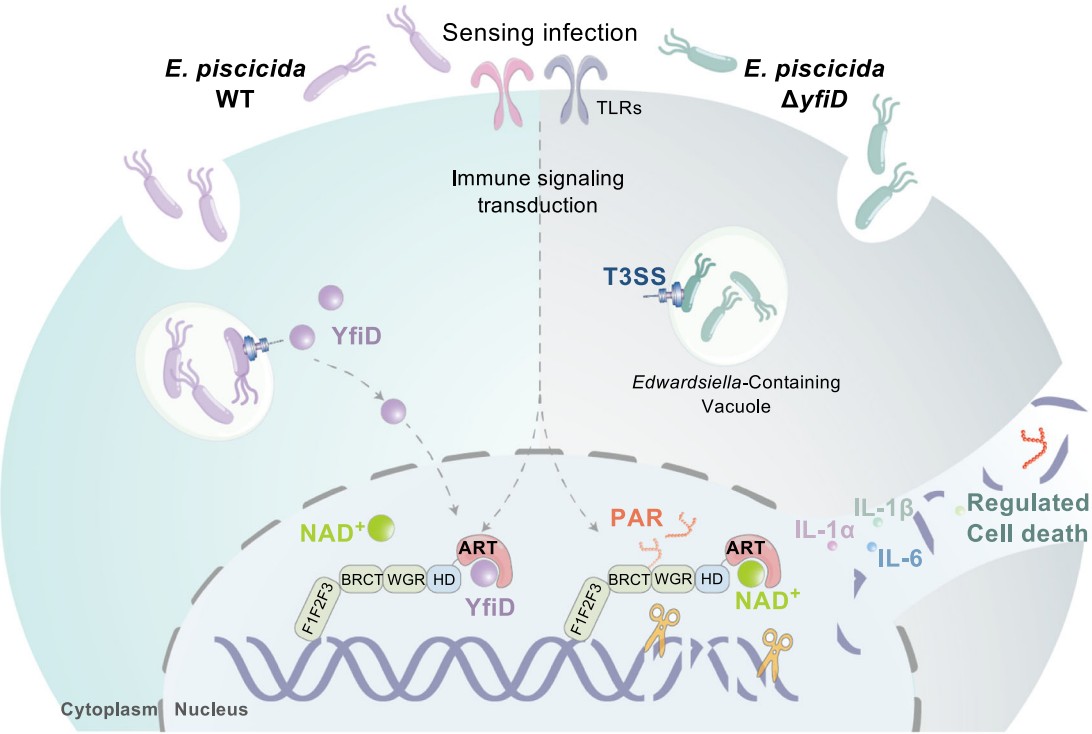

**Fig. 6 Proposed model for the role of YfiD in the pathogenesis of *E. piscicida*.** Upon infection, internalized bacteria-caused DNA lesion is recognized by PARP1 followed by its self-activation. The accumulated PAR leads to enhanced inflammatory response and regulated cell death in the residing macrophages, restraining *E. piscicida*'s replication in macrophages. Along with the occurrence of DNA damage events, YfiD would be delivered into host cells through T3SS and then translocated into the nucleus, where it interacts with the ART domain of PARP1 to impair PAR formation. This blockade rescues *E. piscicida* from niche disruption and inflammatory elimination, facilitating *E. piscicida* proliferation and pathogenicity in the host.

For the intracellular proliferation assay, cells seeded in 24-well plates were infected with *E. piscicida* strains. At the indicated time, the lysates of cells lysed by 1% Triton X-100 solved in sterile PBS were diluted and plated on DHL plates. The amount of bacteria colonies was counted following incubation at 37°C overnight.

For the TEM-1 β-lactamase reporter assay, cells plated in 24-well plates were infected with *E. piscicida* strains carrying the YfiD-TEM1 fusion backboned by pCX340[4]. Briefly, at 6 hpi, cells were incubated with CCF2-AM for another two hours, followed by confocal imaging with an excitation wavelength of 405 nm and emission wavelength of 440-460 nm and 510-530 nm. The substrate CCF2 consists of a cephalosporin core linking coumarin and fluorescein. With an excitation of 405 nm, the coumarin emits blue fluorescence at 447 nm, which serves as the excitation for the fluorescein based on fluorescence resonance energy transfer (FRET). As a result, the intact CCF2 emits a green fluorescence signal (at 518 nm). In the presence of β-lactamase (TEM-1 in this assay), the cleavage of the β-lactam ring of the cephalosporin disrupts FRET and results in an emission of 447 nm (Blue) (Fig. S1c).

**Protein expression and purification**. Coding sequences of indicated proteins were subcloned into 6×His-tagged pET28a digested by *Nco*I and *Xho*I. *E. coli* BL21 (DE3) strains containing the corresponding recombinant plasmids were cultured in 300 mL LB medium at 37°C until the logarithmic phase. Subsequently, *E. coli* cells were maintained with 0.2 mM IPTG at 18°C for 12-16 h. Bacterial cultures were harvested through centrifugation and resuspended in lysis buffer (20 mM Tris, 150 mM NaCl, adjusted pH). Clarified supernatant acquired from high-pressure homogenization was purified via His-tag purification resin (Beyotime)-

mediated affinity purification and eluted with lysis buffer containing 500 mM imidazole, which was removed through dialysis in lysis buffer supplemented with 5% glycerol and stored at -80°C.

**Western blot assay**. Protein samples from monolayer cells lysed with SDS lysis buffer (Beyotime) and cultured bacteria harvested by centrifugation were heated at 95°C for 10 min, followed by denaturation with protein loading buffer (YEASEN). The proteins were separated by denaturing polyacrylamide gel electrophoresis (SDS-PAGE) and transferred to a PVDF (polyvinylidene fluoride) membrane activated with methanol. The membrane was then blocked with 5% skim milk (Beyotime) dissolved in PBST (PBS containing 0.05% Tween-20) at room temperature for 2 h and then incubated with appropriate primary antibody at 4 °C overnight. After washing three times with PBST, the membrane was incubated with the secondary antibody conjugated to horse radish peroxidase at room temperature for 2 h. Following the washing described above, an ECL detection kit (Epizyme) was used to visualize immunoreactions.

The relative density of targeted proteins was calculated by gray scanning in Image J. Briefly, the obtained images were transformed to an 8-bit format and removed from the background. Following inverted, the density of each band was measured.

**Pull-down assay**. To screen out YfiD-interacting proteins in HeLa cells, $8×10^7$ HeLa-YfiD-HA or HeLa-Mock were washed with pre-cold PBS and lysed with cell lysis buffer (20 mM Tris, 150 mM NaCl, 1% Triton X-100 and protease inhibitors, Beyotime). The supernatant of lysates was acquired through centrifugation and rotated with 200 μL anti-HA magnetic beads (Beyotime) at 4 °C overnight. Beads were washed several times

with TBS (Beyotime) until $OD_{280} < 0.05$ and eluted with protein loading buffer (YEASEN) heating at 95℃ for 10 min. The obtained immunoprecipitates were separated by SDS-PAGE and identified by mass spectrometric analysis.

To investigate the interaction between YfiD and the ART domain of PARP1, the fragment containing 6×His-GFP, 6×His-GFP-YfiD, or S-tagged ART fusion was cloned into plasmid pETDuet-1, respectively. *E. coli* BL21 (DE3) strains containing the corresponding recombinant plasmids were cultured in 100 mL LB medium at 37 ℃ until reaching the logarithmic phase. Subsequently, *E. coli* cells were maintained with 0.5 mM IPTG at 25 ℃ for 6 h. Bacterial cultures were harvested through centrifugation and resuspended in 20 mL lysis buffer (20 mM Tris, 150 mM NaCl, pH 8.0). The clarified supernatant obtained from high-pressure homogenization (1 mL) containing 6×His-GFP or 6×His-GFP-YfiD was incubated with the supernatant containing S-tagged ART (1 mL) and 30 μl protein A beads (Beyotime) at room temperature for 1 h. The separated supernatant (Input) was rotated with 30 μl anti-GFP magnetic beads (AlpaLifeBio) at room temperature for 2 h. Subsequently, beads were washed three times with lysis buffer and eluted with 70 μl glycine-elution buffer (0.2 M glycine, pH 2.5) at room temperature for 10 min. The eluted sample was neutralized with 35 μl neutralization buffer (1 M Tris, pH 10.4) and analyzed by immunoblotting.

**Mass spectrometry analysis**. MS/MS analysis was carried out by APTBIO (Shanghai, China). Briefly, protein bands were cut and incubated with destain buffer (30% acetonitrile, 100 mM $NH_4HCO_3$) at 37 ℃ until the gels were destained. After destaining, the gel pieces were digested with trypsin at 37℃ overnight. Liquid chromatography-mass spectrometry (LC-MS) analyses of peptides were performed with a Q Exactive mass spectrometer based on higher-energy collisional dissociation (HCD). Mass spectrometric data were analyzed using the Mascot 2.2 software for database search.

**Bacterial adenylate cyclase two-hybrid (BACTH) system**. Coding sequences of YfiD and single domain of PARP1 were fused to fragment T18 in pUT18 and fragment T25 in pKT25, respectively. The generated recombinant plasmids were co-transformed into competent BTH101 cells and then plated on LB plates supplemented with 100 μg/mL Carb, 50 μg/mL Km, 0.5 mM IPTG, and 40 μg/mL X-Gal. After incubation at 37℃ for 24-72 h, the appearance of bacterial colonies was photoed to determine the presence of protein-protein interaction (blue) or not (white).

For ONPG-based colorimetric analysis, pairwise transformants were grown in 5 mL LB medium supplemented with 100 μg/mL Carb, 50 μg/mL Km, and 0.5 mM IPTG overnight. Subsequently, 700 μL Z-buffer (60 mM $Na_2HPO_4$, 40 mM $NaH_2PO_4$, 10 mM KCl, and 10 mM $MgSO_4$ at pH 7.0), 30 μL trichloromethane, and 30 μL 0.1% SDS were mixed thoroughly with 200 μL bacterial pellets. A total of 200 μL ONPG (4 mg/mL) was added followed by incubation at 30℃. Once sufficient color occurred, 1 M sodium carbonate was added to stop the reaction, and the precise time was noted. Bacterial pellets were discarded by centrifugation and $OD_{420}$ values were measured. The final β-galactosidase activity was calculated according to the formula: $1000 \times OD_{420}/(OD_{600} \times 0.2 \, mL \times min)$.

**PARP1 chemiluminescent assay**. To explore the influence on PARP1 activity by YfiD, the PARP1 chemiluminescent assay was performed and the corresponding kit was purchased from BPS Bioscience. Briefly, the enzymatic activity of purified PARP1 was quantified with histone substrates, activated DNA, and tested inhibitors. Biotinylated $NAD^+$ was used as the substrate of PARP1-catalysed PARylation. Biotin-streptavidin interaction and streptavidin-HRP (Horseradish peroxidase) complex allow the chemiluminescent detection of PAR. Initially, histone mixtures were coated on a 96-well plate overnight. Then, biotinylated $NAD^+$, activated DNA, and tested inhibitors were incubated with purified PARP1 to initiate PARylation of histone proteins. Finally, the plate was treated with streptavidin-HRP followed by the addition of ECL substrate to produce chemiluminescence that can be measured using a chemiluminescence reader.

**Lentivirus-mediated stable expression system**. The open reading frame of YfiD-HA was cloned into pCDH vector and verified by sequence. HEK293T cells seeded in a 10 cm dish overnight were transfected using Vigofect (Vigorous Biotechnology)-mediated plasmid delivery (pCDH-YfiD-HA, pCMV-VSVG, and pCAG-dR8.9) to support lentiviral packaging according to the manufacturer's instructions. Recombinant virus particles were harvested using Lenti-X Concentrator (Takara) and infected with HeLa or J774A.1 cells plated in 6-well plates. At 48 hpi, the culture supernatant was removed and cells expressing YfiD-HA were selected with an appropriate concentration of puromycin (HeLa: 1.5 μg/mL, J774: 3 μg/mL). The corrected cell lines were confirmed by Western blot with an anti-HA primary antibody.

**PI uptake assay**. At the indicated time following reagents treatment or *E. piscicida*'s infection, cells prepared in 24-well plates were stained with 10 μg/mL propidium iodide (PI) in a 37 ℃ incubator containing 5% $CO_2$ for 10 min. Images were then acquired by a fluorescence microscope with an excitation wavelength of 561 nm and a bright field, respectively.

**In vivo bacterial infection assays**. For bacterial colonization in vivo, a total of 10 turbots (*Scophthalmus maximus*, 30 g per fish) were injected intraperitoneally with indicated strains at a dose of $1 \times 10^6$ CFUs/fish. At 8 days postinfection, turbots were sacrificed and the organs, including the liver, spleen, and kidney, were collected. Bacterial burdens were determined by plating dilutions of organ homogenates with appropriate antibiotics.

For survival curves of zebrafish (*Danio rerio*, 3 cm per fish) challenged with *E. piscicida*, a total of 20 zebrafish were intramuscularly injected with indicated strains at a dose of 400 CFUs/fish or PBS as negative controls. The mortality of fish was recorded for 7 days post-infection. The infection experiments were performed in at least three independent replicates.

**Protein accession numbers**. YfiD (D0ZD05) and PARP1 (Q921K2) are obtained from UniProt. IL-1α (NP_034684.2), IL-1β (NP_032387), and IL-6 (NP_112445.1) are acquired from NCBI database.

**Statistics and reproducibility**. Data were presented as the mean ± S.D. of triplicate samples per experimental condition unless noted otherwise. Representative results are shown in the figures. Statistical evaluation was performed using paired or unpaired Student's *t*-test. Data were shown as mean ± SD, and *$P < 0.05$, **$P < 0.01$, and ***$P < 0.001$ was considered significant.

**Reporting summary**. Further information on research design is available in the Nature Portfolio Reporting Summary linked to this article.

## Data availability

The mass spectrometry data has been deposited in PRIDE with identifiers PXD047656 and PXD048313. Microscope images and column-related data are available in Supplementary data 1-2. Plasmids and strains generated in this study are available from the corresponding author upon reasonable request. The complete sequences of the corresponding plasmids have been deposited in NCBI under the accession number PP098725-PP098733.

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

## Acknowledgements

This work was supported by grants from the National Natural Science Foundation of China (32373183, 32130108), the National Key Research and Development Program (2022YFE0101200), the Shanghai Municipal Science and Technology Commission Technical Leader Project (21XD1431900), and the China Agriculture Research System of MOF and MARA (CARS-47).

## Author contributions

Conceptualization: SS QYW; Data curation: MQZ SS; Funding acquisition: SHC QYW; Investigation: MQZ YBL SS; Project administration: YXZ; Supervision: YBZ YM SS; Validation: SS QYW; Visualization: MQZ SS; Writing – original draft: MQZ SS; Writing – review and editing: SS.

## Competing interests

The authors declare no competing interests.

## Ethics

We have complied with all relevant ethical regulations for animal use. All animal procedures performed were authorized by the animal care committee of the East China University of Science and Technology (2006272). The Experimental Animal Care and Use Guidelines from the Ministry of Science and Technology of China (MOST-2011-02) were rigorously adhered to.
