## [Peer Review File · Communications Biology]

Reviewers' comments:

Reviewer #1 (Remarks to the Author):

In this study, Mengqing Zhou et al., have identified YfiD as a bacterial T3SS effector that specifically blocks PARP1 activation and hence the PARylation of itself, and acts as potential bacterial toxin inhibiting PARP1 mediated inflammatory cell death called as parthanatos, and YfiD-dependent augmentation of bacterial replication by protecting the replication niches. Moreover, this study also establishes the specific interaction between the bacterial effector YfiD and the host protein PARP1 and that PARP1 inhibitors may be helpful to fight bacterial infections such as *E. piscicida*. Overall, the study is designed and executed well, and reported novel findings with clinical relevance. I only have a few minor comments as discussed below.

Minor comments:

1. Recently, the non-developmental pathways of cell death are named as 'regulated cell death' (RCD) modalities, and the 'programmed cell death' (PCD) is exclusively reserved for developmental and homeostatic mechanisms. So, the authors need to revise the manuscript text accordingly.
2. Authors mention PARP1-driven parthanatos as a novel mode of cell death, but it is not. So, correct this error.
3. While it is interesting to find that excessive activation of PARP1 amplifies DNA damage, the Olaparib treatment in Fig3a failed to block the pH2AX levels, contradicting the claim that PARP1 inhibition leads to reduced DNA damage or may act similar to the bacterial PARP1 inhibitor YfiD. Moreover, the changes in pH2AX levels in Fig3C/D appear to be marginal at best. However, the role of YfiD in reducing the PARP1 activation and PAR levels is fully supported by the data.
4. Provide the CFU data corresponding to the Z-VAD-FMK treated samples from Fig3A, which will help addressing if addition of Z-VAD-FMK switches the cell death to necroptosis and hence may still reduce the bacterial replication.

Reviewer #2 (Remarks to the Author):

In this manuscript, Zhou and co-authors identified a new factor in the proteome of *Edwardsiella piscicida*, able to inhibit the catalytic activity of PARP1, and to interfere with the pathogenesis of the infection through the inhibition of the PARP1 induced cell death. The manuscript is clear and overall convincing, and the findings are interesting for a broad audience, since they have implications in different fields (i.e., pathogen-host interaction in human and animal infections, PARP1 inhibition in human diseases). Nevertheless, some points need to be fully addressed with further experimental work.

1. The experiment in Figure 1B is poorly described. The authors state that the YfiD protein was probed with an anti-flag antibody, but it is not described how the flag-tagged construct was obtained and transfected into bacteria. In addition, a loading control for supernatant sample is missing. The molecular weight markers are missing.
2. In the Figure 1C the IF images are of poor quality. To clearly show the percentage of blue cells in the total cells it could be of help to show the IF channels separately. In addition, it is not clear what the green signal represents (stain, autofluorescence?). The green signal was shown to mark all the cells, but it seems to vary a lot of intensity among different cells. Perhaps, to show all the cells, the

authors could acquire brightfield images of the same fields, compared to IF blue channel.

3. In the Figure 2D the authors report a two-hybrid experiment to assess interaction between PARP1 domains and YfiD, but, in this assay, they failed to restore interaction between the full-length PARP1 protein and YfiD. A clear demonstration of physical interaction between the two proteins is essential for the manuscript to be fully convincing. In this regard I would suggest performing a pull-down assay with recombinant proteins.

4. In Figures 3 and 4 the molecular weight markers are missing.

Reviewer #3 (Remarks to the Author):

Brief summary of the manuscript:

The authors generally seem to have made some interesting discoveries, identifying a new protein (YfiD) of *Edwardsiella piscicida*, its effects on PARylation and the consequences of its presence/absence for cell survival following DNA damage induction or bacterial infection. I am not convinced that all of the conclusions are entirely accurate or directly supported by the data. Particularly, the combination of DNA damage- and a bacterial infection-related data in my eyes does not ultimately prove the role of YfiD in parthanatos (particularly given the negative MIF and AIF data) and so the role of YfiD for tolerance of *E. piscicida* and cell death should be discussed in more general terms.

Overall impression:

The manuscript presents some insightful data, but some of the conclusions drawn seems inaccurate or at least not fully supported by the data. The authors seem set on a certain mechanism (YfiD inhibiting PARP1 and thus preventing a specific type of cell death – parthanatos), but a more open discussion on the picture that the different pieces of data paint is needed. The presentation of some of the data also needs improving (.e.g. molecular weight markers, missing controls), some more of the already existing data should be provided (MS data, data/images of TEM 1 beta-lactamase report assay) and some further experimentation is suggested (see below with respect to Figures 2 and 5).

Specific comments:

Abstract / author summary / introduction:

In the abstract the authors set out what they believe to be the main findings, namely a) the discovery of a new protein, DfiY, as a T3SS effector, b) its function as a PARP1 inhibitor and c) role in preventing pathanatos in response to bacterial infection by *E. piscicida*. I think it is better to refrain here from talking about parthanatos and PARP1 hyperactivation, as this is not definitely shown in the data. The same applies to the author summary and introduction.

Results (Figure 1):

The highlighted band in Figure 1A does not quite correspond to 14 kDa as per the marker, but this might be the result of resolution of low molecular weight fractions at the bottom of the gel or uneven running of the gel. The actual MS data showing the fractions corresponding to YfiD should be included

in Figure S1. The result in Figure 1B looks quite interesting, but it would be good to include the molecular weight markers and briefly discuss why the deltaT3SS strain expresses a lot more of YfiD and the deltaT6SS strain exports more YfiD than the WT strain. The result in Figure 1D is quite striking, but the representative images do not reflect that. More images should be provided to make the purported effect in Figure 1D believable. It would have also been good to include controls of EseG translocation from the T3SS and the deltaT6SS strains, providing further evidence that YfiD is a T3SS effector.

Results (Figure 2):

Again, the results in Figure 2A and B seem interesting, but the MS data showing the identified unique peptide of PARP1 (and maybe also the other listed proteins) should be included as supplementary figures at least. Figure S2A would profit from including the molecular weight markers and a bigger section of the HA gel to show that no protein fragments are expressed (same goes for Figure S2B). Figure 2 again looks interesting, but a negative control with Zip could have been included (e.g. Zip and ART or Zip and Full). The conclusion on lines 162-163 seems wrong. If YfiD binds the catalytic site, then binding of YfiD to inactive PARP1 is not absent due to misfolding of PARP1 in *E. coli*, but because the HD domain covers the catalytic site in inactive PARP1. A HD-ART polypeptide could have been tested in Figure 2D for further evidence. Conclusions about the binding characteristics of YfiD to PARP1 could have further profited from testing ART variants carrying different mutations of residues in the active site. Similarly, YfiD fragments could have been created to assess which part of the protein binds PARP1 and some analysis of the amino acid sequence carried out to identify motifs/domains. The results in Figure 2F look interesting. Details of the chemiluminescent assay should be provided (in the Materials & Methods section at least).

Results (Figure 3):

The word "excessive" on line 175 should be removed and line 183 reworded to say that "...PARylation was suppressed by the PARP inhibitor Olaparib...". The results in Figure 3A look clean, but again should include the molecular marker and could even profit from showing more of the PARylation profile (e.g. to see histone ADP-ribosylation). Lines 192-193 should be reworded to relate more generally to the role PARP1 activation and PAR formation in the inflammatory response (rather than the restraint of *E. piscicida* intracellular survival specifically, or provide the reference that shows this mechanism). Again, the results in Figures 3C and E are interesting, but should include molecular markers. In Figure 3C the deltaYfiD mutant is missing the 0 hr time point. Figure 3D and F are a useful visualization of the results in Figures 3C and E, but the position of the legends at the top of the figures is slightly confusion. They could be moved over and instead lines introduced to highlight which bars belong to which of the WT/mock and deltaYfiD/YfiD. Figure 3B should be removed as this data seems to be also present in Figure 5B.

Results (Figure 4):

The results in Figure 4A and B look interesting. Figure 4B could be improved by providing the degree to which the difference between Z-VAD-FMK alone or together with YfiD at 6hr and 8hr is statistically significant. YfiD recruitment in Figure 4C is interesting, but the data with regards to MIF and AIF is in disagreement with the parthanatos hypothesis and the manuscript does not provide any evidence of how a MIF- and AIF-independent parthanatos could look like. The recruitment dynamics of YfiD are also different to its purported role in suppressing PARylation following *E. piscicida* infection (no time frame provided) or DNA damage induction (shorter time frame than YfiD nuclear localization). This section of the results should be re-worded to move away from parthanatos to talking about cell death in general. Discussing two different reported aspects of YfiD function, namely suppression of PAR in

early DNA damage/infection response (again, time frame of PAR formation in infection response needs to be clarified) and later YfiD recruitment to the nucleus, here could be useful. A PI uptake assay does not allow a conclusion as to the type of cell death that occurs, so the term "necrotic" on line 231 should also be removed.

Results (Figure 5):

As elsewhere, the results seem interesting but require some re-wording and/or work with regards to the figures. The number "20" on the y-axis in Figure 5A seems to be somewhat misplaced. IL-1alpha, IL-1beta and IL-6 are secreted cytokines, so the conclusion that a PAR-accelerated proinflammatory response promotes clearance of intracellular *E. piscicida* seems not quite right. The bacterial infection assay does not provide how many cells survived under the different conditions, so instead a conclusion from Figure 5B could also be that cell death (rather than intracellular clearance) occurs at a lower intracellular level of deltaxfiD vs. WT *E. piscicida* and so surviving cells infected with deltaxfiD *E. piscicida* have a lower bacterial count. If available, providing the percentage of surviving cells under the different conditions would be useful in this context.

The transition from the conclusions in Figure 5A and B to Figure 5C and D needs to be better explained. Is it that cell death at a lower intracellular level of deltaxfiD compared to WT *E. piscicida* results in lower bacterial burden in different organs and increased survival of the organism? This seems to conflict with the notion that YfiD inhibits PARP1-directed PARylation and inflammatory response following infection. Should that not increase the tolerance of the organism to infection and allow better survival (and more time for the bacteria to propagate), or does it lead to a rapidly higher *E. piscicida* titer in the organism and faster death? This needs to be discussed. Helpful in this context would be to provide a PI uptake assay similar to that in Figure 4A, but measuring cell death in response to deltaxfiD and WT *E. piscicida* infection under the different conditions used in Figure 4A. The term "programmed" on line 288 and the term "proinflammatory" on line 291 should be removed as no definitive conclusions on the type of cell death can be made.

Discussion:

The discussion needs to be re-written and the focus on "parthanatos" (e.g. line 300) and "necrosis" (line 304, conflicting term) removed and instead cell death in general discussed. The observations with regards to the immediate PARP1 response to infection *E. piscicida* and subsequent events (increase in expression in cytokine genes, differences in intracellular *E. piscicida* levels) need to be separated out in the discussion or at least discussed how the different data fit together, since it cannot conclusively be said that suppression of early PARylation causes the later events (for example, Figure 3E does not specify when after infection the reported PARylation pattern was observed and how that relates to the timing of the nuclear recruitment of YfiD in Figure 4C).

Reviewers' comments:

Reviewer #1 (Remarks to the Author):

In this study, Mengqing Zhou et al., have identified YfiD as a bacterial T3SS effector that specifically blocks PARP1 activation and hence the PARylation of itself, and acts as potential bacterial toxin inhibiting PARP1 mediated inflammatory cell death called as parthanatos, and YfiD-dependent augmentation of bacterial replication by protecting the replication niches. Moreover, this study also establishes the specific interaction between the bacterial effector YfiD and the host protein PARP1 and that PARP1 inhibitors may be helpful to fight bacterial infections such as *E. piscicida*. Overall, the study is designed and executed well, and reported novel findings with clinical relevance. I only have a few minor comments as discussed below.

Minor comments:

1. Recently, the non-developmental pathways of cell death are named as 'regulated cell death' (RCD) modalities, and the 'programmed cell death' (PCD) is exclusively reserved for developmental and homeostatic mechanisms. So, the authors need to revise the manuscript text accordingly.

Response: Thank you for your professional suggestion. We have accordingly modified the manuscript.

2. Authors mention PARP1-driven parthanatos as a novel mode of cell death, but it is not. So, correct this error.

Response: We sincerely appreciate the valuable correction and this error has been corrected.

3. While it is interesting to find that excessive activation of PARP1 amplifies DNA damage, the Olaparib treatment in Fig3a failed to block the pH2AX levels, contradicting the claim that PARP1 inhibition leads to reduced DNA damage or may act similar to the bacterial PARP1 inhibitor YfiD. Moreover, the changes in pH2AX

levels in Fig3C/D appear to be marginal at best. However, the role of YfiD in reducing the PARP1 activation and PAR levels is fully supported by the data.

Response: Thank you for your constructive comment. This conclusion “excessive activation of PARP1 amplifies DNA damage” was not claimed in the manuscript. Here, we emphasized that YfiD inhibits the PARP1 activation and impairs PAR accumulation, which may potentially reduce DNA damage events. In fact, specific conditions, such as MNNG treatment or bacterial infection, can produce DNA damage events which further lead to PARP1 excessive activation. The excessive activation of PARP1 may result in the depletion of NAD⁺ and interfere with other parallel DNA repair pathways. Olaparib, a PARP1 inhibitor, can block PARP1 excessive activation but not the generation of DNA damage. Our results indicated that bacteria-delivered or stable-expressed YfiD can slightly relieve DNA damage caused by infection or DNA alkylation, respectively.

4. Provide the CFU data corresponding to the Z-VAD-FMK treated samples from Fig3A, which will help addressing if addition of Z-VAD-FMK switches the cell death to necroptosis and hence may still reduce the bacterial replication.

Response: Thank you for your constructive suggestion. In the original manuscript, Figure 3B and Figure 5B shared some repetitive data, therefore, Figure 3B has been removed. To explore the effect of Z-VAD-FMK on *E. piscicida*'s replication in macrophages, the CFU of *E. piscicida* WT in J774A.1 was determined in the presence and absence of Z-VAD-FMK. The result showed that Z-VAD-FMK treatment slightly reduced *E. piscicida*'s replication in macrophages. Given that the cell death pathways in *E. piscicida*-infected cells remain obscure, more research needs to be performed in the future.

Reviewer #2 (Remarks to the Author):

In this manuscript, Zhou and co-authors identified a new factor in the proteome of *Edwardsiella piscicida*, able to inhibit the catalytic activity of PARP1, and to interfere with the pathogenesis of the infection through the inhibition of the PARP1 induced cell death. The manuscript is clear and overall convincing, and the findings are interesting for a broad audience, since they have implications in different fields (i.e., pathogen-host interaction in human and animal infections, PARP1 inhibition in human diseases). Nevertheless, some points need to be fully addressed with further experimental work.

1. The experiment in Figure 1B is poorly described. The authors state that the YfiD protein was probed with an anti-flag antibody, but it is not described how the flag-tagged construct was obtained and transfected into bacteria. In addition, a loading control for supernatant sample is missing. The molecular weight markers are missing.

Response: Thank you for your kind suggestions. The detailed information on the flag-tagged construction and the transformation into bacteria have been added to the manuscript in “Materials and methods” (Lines 421-424).

In Figure 1B, two loading controls of EseB and EvpP are included now. EseB is the secreted apparatus of T3SS and EvpP is a secreted effector depending on T6SS. The molecular weight markers have been added.

2. In the Figure 1C the IF images are of poor quality. To clearly show the percentage of blue cells in the total cells it could be of help to show the IF channels separately. In addition, it is not clear what the green signal represents (stain, autofluorescence?). The green signal was shown to mark all the cells, but it seems to vary a lot of intensity among different cells. Perhaps, to show all the cells, the authors could acquire brightfield images of the same fields, compared to IF blue channel.

Response: Thank you for your constructive suggestion. To obtain IF images of higher quality, we optimized several conditions, including the removal of antibiotics and cell densities. The newly captured images are shown in Figure 1C. To make it clear, the separate IF channels have been demonstrated.

The detailed description of TEM-1 (β -lactamase) reporter assay has also been added in “Materials and methods” (Lines 437-448) and Figure S1C. Specifically, upon the cleavage by cytoplasmic esterases, CCF2 would emit green fluorescence signals

with an excitation of 409 nm, which represent all living cells (See the two images below). The differences in cell intensity may be attributed to the uneven distribution of cells, which has been improved (See Figure 1C). When TEM-1 is transported with the effector, the cleavage of the β -lactam ring of the cephalosporin results in an emission of 447 nm and produces blue fluorescence signals, which indicates the localization of the effector. In general, the results generated by TEM-1 β -lactamase reporter assay are present as Fig 1C and D, including the ratio and the corresponding images (See references 1-2 below).

1. Xu Y, Zhou P, Cheng S, Lu QH, Nowak K, Hopp AK, Li L, Shi XY, Zhou ZW, Gao WQ, Li D, He HB, Liu XY, Ding JJ, Hottiger MO, Shao F. 2019. A bacterial effector reveals the V-ATPase-ATG16L1 axis that initiates xenophagy. *Cell* 178:552-566.
2. Chen H, Yang DH, Han FJ, Tan JC, Zhang LZ, Xiao JF, Zhang YX, Liu Q. 2017. The bacterial T6SS effector EvpP prevents NLRP3 inflammasome activation by inhibiting the Ca^{2+} -dependent MAPK-Jnk pathway. *Cell Host Microbe* 21:47-58.

3. In the Figure 2D the authors report a two-hybrid experiment to assess interaction between PARP1 domains and YfiD, but, in this assay, they failed to restore interaction between the full-length PARP1 protein and YfiD. A clear demonstration of physical interaction between the two proteins is essential for the manuscript to be fully convincing. In this regard I would suggest performing a pull-down assay with recombinant proteins.

Response: Thank you for your enlightening suggestion. In the original manuscript, we speculated that the failed interaction between the full-length PARP1 protein and YfiD would be probably due to the improper folding of PARP1 expressed in *E. coli*. Reviewer #3 proposed a more reasonable possibility that HD domain might cover the

catalytic site and thus inactive PARP1. To test this hypothesis, the interaction between HD-ART polypeptide and YfiD was tested through a bacterial two-hybrid assay. The disappeared interaction between HD-ART and YfiD indicated that the HD domain indeed prevents the ART domain from contacting YfiD (Figure 2D-E, Line 184-187). This inhibitory effect of HD could explain the failed interaction between the full-length PARP1 protein and YfiD. To further verify the interaction between YfiD and ART, a GFP beads-mediated pull-down assay was performed (Figure 2F, Line 187-189).

4. In Figures 3 and 4 the molecular weight markers are missing.

Response: Thanks for your kind suggestion. The molecular weight markers have been added.

Reviewer #3 (Remarks to the Author):

Brief summary of the manuscript:

The authors generally seem to have made some interesting discoveries, identifying a new protein (YfiD) of *Edwardsiella piscicida*, its effects on PARylation and the consequences of its presence/absence for cell survival following DNA damage induction or bacterial infection. I am not convinced that all of the conclusions are entirely accurate or directly supported by the data. Particularly, the combination of DNA damage- and a bacterial infection-related data in my eyes does not ultimately prove the role of YfiD in parthanatos (particularly given the negative MIF and AIF data) and so the role of YfiD for tolerance of *E. piscicida* and cell death should be discussed in more general terms.

Overall impression:

The manuscript presents some insightful data, but some of the conclusions drawn seems inaccurate or at least not fully supported by the data. The authors seem set on a certain mechanism (YfiD inhibiting PARP1 and thus preventing a specific type of cell death – parthanatos), but a more open discussion on the picture that the different pieces of data

paint is needed. The presentation of some of the data also needs improving (.e.g. molecular weight markers, missing controls), some more of the already existing data should be provided (MS data, data/images of TEM 1 beta-lactamase report assay) and some further experimentation is suggested (see below with respect to Figures 2 and 5).

Specific comments:

Abstract / author summary / introduction:

In the abstract the authors set out what they believe to be the main findings, namely a) the discovery of a new protein, YfiD, as a T3SS effector, b) its function as a PARP1 inhibitor and c) role in preventing pathanatos in response to bacterial infection by *E. piscicida*. I think it is better to refrain here from talking about parthanatos and PARP1 hyperactivation, as this is not definitely shown in the data. The same applies to the author summary and introduction.

Response: Thank you for pointing this out. We have revised our manuscript throughout the whole manuscript accordingly.

Results (Figure 1):

The highlighted band in Figure 1A does not quite correspond to 14 kDa as per the marker, but this might be the result of resolution of low molecular weight fractions at the bottom of the gel or uneven running of the gel. The actual MS data showing the fractions corresponding to YfiD should be included in Figure S1. The result in Figure 1B looks quite interesting, but it would be good to include the molecular weight markers and briefly discuss why the deltaT3SS strain expresses a lot more of YfiD and the deltaT6SS strain exports more YfiD than the WT strain. The result in Figure 1D is quite striking, but the representative images do not reflect that. More images should be provided to make the purported effect in Figure 1D believable. It would have also been good to include controls of EseG translocation from the T3SS and the deltaT6SS strains, providing further evidence that YfiD is a T3SS effector.

Response: Thank you for your kind suggestion. As you mentioned, the highlighted band in Figure 1A does not resemble 14 kDa indicated by the protein marker probably due to the low resolution of small molecular weight proteins. To this end, the MS data of the representative unique peptide corresponding to YfiD has been added in Figure S1B.

In Figure 1B, the molecular weight markers have been added. YfiD expression and secretion features have been discussed in Lines 130-134. In Figure 1C, several conditions, including the removal of antibiotics and cell densities were optimized to obtain IF images with higher quality. Meanwhile, EseG translocation profiles in the $\Delta T3SS$ and $\Delta T6SS$ strains have been included and the blue and green channels were separated to make them more clarified.

Results (Figure 2):

Again, the results in Figure 2A and B seem interesting, but the MS data showing the identified unique peptide of PARP1 (and maybe also the other listed proteins) should be included as supplementary figures at least. Figure S2A would profit from including the molecular weight markers and a bigger section of the HA gel to show that no protein fragments are expressed (same goes for Figure S2B). Figure 2 again looks interesting, but a negative control with Zip could have been included (e.g. Zip and ART or Zip and Full). The conclusion on lines 162-163 seems wrong. If YfiD binds the catalytic site, then binding of YfiD to inactive PARP1 is not absent due to misfolding of PARP1 in *E. coli*, but because the HD domain covers the catalytic site in inactive PARP1. A HD-ART polypeptide could have been tested in Figure 2D for further evidence. Conclusions about the binding characteristics of YfiD to PARP1 could have further profited from testing ART variants carrying different mutations of residues in the active site. Similarly, YfiD fragments could have been created to assess which part of the protein binds PARP1 and some analysis of the amino acid sequence carried out to identify motifs/domains. The results in Figure 2F look interesting. Details of the chemiluminescent assay should be provided (in the Materials & Methods section at least).

Response: Thank you for your enlightening suggestion. The MS data showing the representative of identified unique peptides of PARP1 has been included in Figure S2B.

In Figure S2A and Figure S2F (referred to the previous Figure S2B), the molecular weight markers have been added, and the bigger section of the HA gels were shown. In Figure 2D-E, negative controls of Zip-YfiD and ART-Zip have been included. Details of the chemiluminescent assay have been provided in “Materials and methods” in Lines 525-536.

As mentioned in the previous response, the HD domain blocks the interaction between YfiD and ART, supported by the negative interaction between HD-ART and YfiD. Negative controls of Zip-YfiD and ART-Zip have been included (Figure 2D-E, Lines 184-187). Moreover, several HD-ART variants carrying different mutations in HD were tested as well (Lines 190-203, Figure 2G-H and Figure S2C). The folded HD, consisting of α B, α D, and α F, has been identified to be indispensable for blocking the interaction between ART and YfiD.

Results (Figure 3):

The word “excessive” on line 175 should be removed and line 183 reworded to say that “...PARylation was suppressed by the PARP inhibitor Olaparib...”. The results in Figure 3A look clean, but again should include the molecular marker and could even profit from showing more of the PARylation profile (e.g. to see histone ADP-ribosylation). Lines 192-193 should be reworded to relate more generally to the role PARP1 activation and PAR formation in the inflammatory response (rather than the restraint of *E. piscicida* intracellular survival specifically, or provide the reference that shows this mechanism). Again, the results in Figures 3C and E are interesting, but should include molecular markers. In Figure 3C the *delta*yfiD mutant is missing the 0 hr time point. Figure 3D and F are a useful visualization of the results in Figures 3C and E, but the position of the legends at the top of the figures is slightly confusion. They could be moved over and instead lines introduced to highlight which bars belong to which of the WT/mock and *delta*yfiD/YfiD. Figure 3B should be removed as this data seems to be also present in Figure 5B.

Response: Thank you for your insightful comment. The word “excessive” has been removed (Line 217) and “...PARylation was suppressed by the PARP inhibitor Olaparib...” has been reworded (Line 224-228). In Figure 3A, the molecular markers

have been included. The anti-PAR antibody can recognize poly ADP-ribose and indicate the PARylation profile of all PARylated proteins, including histones, transcription factors, and PARP1 itself. Figure 3B and the relevant description have been removed. The related information has been rephrased in Lines 312-325.

In Figure 3C and E (now 3B and C), the molecular markers have been included. In Figure 3C (now 3B), the uninfected cells at the 0 hr time point served as a control for both cells infected with WT and $\Delta yfiD$. Considering that Figure 3B and C are more intuitive, the previous Figure 3D and F have been removed.

Results (Figure 4):

The results in Figure 4A and B look interesting. Figure 4B could be improved by providing the degree to which the difference between Z-VAD-FMK alone or together with YfiD at 6hr and 8hr is statistically significant. YfiD recruitment in Figure 4C is interesting, but the data with regards to MIF and AIF is in disagreement with the parthanatos hypothesis and the manuscript does not provide any evidence of how a MIF- and AIF-independent parthanatos could look like. The recruitment dynamics of YfiD are also different to its purported role in suppressing PARylation following *E. piscicida* infection (no time frame provided) or DNA damage induction (shorter time frame than YfiD nuclear localization). This section of the results should be re-worded to move away from parthanatos to talking about cell death in general. Discussing two different reported aspects of YfiD function, namely suppression of PAR in early DNA damage/infection response (again, time frame of PAR formation in infection response needs to be clarified) and later YfiD recruitment to the nucleus, here could be useful. A PI uptake assay does not allow a conclusion as to the type of cell death that occurs, so the term “necrotic” on line 231 should also be removed.

Response: Thank you for your enlightening suggestion. In Figure 4B, the statistical analysis related to Z-VAD-FMK treatment has been provided. Parthanatos functions either in an AIF-dependent or in an AIF-independent manner in different cell types. So far, several cell types have been demonstrated to perform the AIF-independent parthanatos caused by pathological stimulations (see references 1-3 below). However, the detailed mechanism involved in the AIF-independent pathway remains obscure,

such as proteins or nucleases. In this manuscript, we focus on the role of YfiD in host-pathogen interaction, especially the interaction between YfiD and PARP1 and its outcome. Therefore, we would lower the tone of AIF-independent parthanatos in the revised manuscript.

To figure out the dynamic localization of YfiD, the distribution characteristic of YfiD was investigated in the early phase. YfiD was transported into the nucleus either with a 15-min MNNG treatment or at 2 hours post- $\Delta yfiD$ -infection (Fig. S2G, Line 289-294). The occurrence of YfiD in the nucleus correlated with the PAR accumulation during the early MNNG treatment or infection stage of *E. piscicida*. The term “necrotic” has been changed to “with the membrane disruption” (Line 265).

References

1. Jang KH, Do YJ, Son D, Son E, Choi JS, Kim E. 2017. AIF-independent parthanatos in the pathogenesis of dry age-related macular degeneration. *Cell Death Dis* 8:e2526.
2. Regdon Z, Robaszkiewicz A, Kovács K, Rygielska Ż, Hegedűs C, Bodoor K, Szabó É, Virág L. 2019. LPS protects macrophages from AIF-independent parthanatos by downregulation of PARP1 expression, induction of SOD2 expression, and a metabolic shift to aerobic glycolysis. *Free Radic Biol Med* 131:184-196.
3. Luan YY, Zhang L, Peng YQ, Li YY, Yin CH. 2022. STING modulates necrotic cell death in CD4 T cells via activation of PARP-1/PAR following acute systemic inflammation. *Int Immunopharmacol* 109:108809.

Results (Figure 5):

As elsewhere, the results seem interesting but require some re-wording and/or work with regards to the figures. The number “20” on the y-axis in Figure 5A seems to be somewhat misplaced. IL-1alpha, IL-1beta and IL-6 are secreted cytokines, so the conclusion that a PAR-accelerated proinflammatory response promotes clearance of intracellular *E. piscicida* seems not quite right. The bacterial infection assay does not provide how many cells survived under the different conditions, so instead a conclusion from Figure 5B could also be that cell death (rather than intracellular clearance) occurs at a lower intracellular level of $\Delta yfiD$ vs. WT *E. piscicida* and so surviving cells infected with $\Delta yfiD$ *E. piscicida* have a lower bacterial count. If available, providing

the percentage of surviving cells under the different conditions would be useful in this context.

The transition from the conclusions in Figure 5A and B to Figure 5C and D needs to be better explained. Is it that cell death at a lower intracellular level of *deltayfiD* compared to WT *E. piscicida* results in lower bacterial burden in different organs and increased survival of the organism? This seems to conflict with the notion that YfiD inhibits PARP1-directed PARylation and inflammatory response following infection. Should that not increase the tolerance of the organism to infection and allow better survival (and more time for the bacteria to propagate), or does it lead to a rapidly higher *E. piscicida* titer in the organism and faster death? This needs to be discussed. Helpful in this context would be to provide a PI uptake assay similar to that in Figure 4A, but measuring cell death in response to *deltayfiD* and WT *E. piscicida* infection under the different conditions used in Figure 4A. The term “programmed” on line 288 and the term “proinflammatory” on line 291 should be removed as no definitive conclusions on the type of cell death can be made.

Response: We deeply appreciate your suggestion. In Figure 5A, the number “20” on the y-axis has been aligned. The term “programmed” on line 288 (now Line 332) and the term “proinflammatory” on line 291 have been reworded (Lines 331-334).

Indeed, IL-1 α , IL-1 β , and IL-6 are secreted cytokines. Once secreted to the supernatant, these cytokines can function as signals to act on the cell populations of macrophages (see reference 1 below). We reworded the related description and changed “inflammatory clearance” to “inflammatory response” and did not claim that intracellular IL-1 α , IL-1 β , and IL-6 directly mediated the clearance of intracellular *E. piscicida*. In our proposed model, these cytokines might serve as cell death signals or inducers of inflammatory pathways, which indirectly promote bacterial clearance.

In addition, the PI uptake assay was performed to investigate the survival rates of cells infected with WT and $\Delta yfiD$ as suggested by the reviewer (Figure 5B, Figure S2H, Line 316-321). Consistent with stable-expressed YfiD (Figure 4A-B), *E. piscicida*-delivered YfiD functions to inhibit PARP1-dependent cell death. This result implied YfiD suppresses PARP1-dependent cell death to facilitate the intracellular survival of *E. piscicida* in macrophages.

The infection process of *E. piscicida* contains six steps as illustrated below (see

reference 2 below). Once engulfed into macrophages (Step 5), *E. piscicida* resides in the vacuoles, named *Edwardsiella*-contained vacuoles, to replicate and evade the immune killing. Herein, YfiD plays a role in attenuating infection-induced PAR formation, regulated cell death, and inflammatory response, promoting intracellular bacterial replication (Figure 3-4 and Figure 5A-C). Due to the lack of sufficient intracellular replication, $\Delta yfiD$ failed to systematically infect the host with an alleviated bacterial burden and lethality to fish (Fig. 5D and E). The related discussion has been rephrased in Lines 364-371.

1. Shapouri-Moghaddam A, Mohammadian S, Vazini H, Taghadosi M, Esmaili SA, Mardani F, Seifi B, Mohammadi A, Afshari JT, Sahebkar A. 2018. Macrophage plasticity, polarization, and function in health and disease. *J Cell Physiol* 233:6425-6440
2. Leung KY, Siame BA, Tenkink BJ, Noort RJ, Mok YK. 2012. *Edwardsiella tarda* - virulence mechanisms of an emerging gastroenteritis pathogen. *Microbes Infect* 14:26-34.

Discussion:

The discussion needs to be re-written and the focus on “parthanatos” (e.g. line 300) and “necrosis” (line 304, conflicting term) removed and instead cell death in general discussed. The observations with regards to the immediate PARP1 response to infection *E. piscicida* and subsequent events (increase in expression in cytokine genes, differences in intracellular *E. piscicida* levels) need to be separated out in the discussion or at least discussed how the different data fit together, since it cannot conclusively be said that suppression of early PARylation causes the later events (for example, Figure

3E does not specify when after infection the reported PARylation pattern was observed and how that relates to the timing of the nuclear recruitment of YfiD in Figure 4C).

Response: We deeply appreciate your suggestion. The discussion has been rephrased accordingly. As mentioned in the previous response, the earlier time of YfiD translocation has been demonstrated. The “parthanatos” -related discussions have been condensed and the regulated cell death in general has been discussed.

Reviewers' comments:

Reviewer #2 (Remarks to the Author):

The authors have addressed all my concerns with further experimental data and by adding the informations required in the materials and methods section.

Reviewer #3 (Remarks to the Author):

The authors have made serious efforts to respond to all points of criticism and the data and manuscript are much improved. Remaining comments:

The representative images in Figure 1C still do not seem to align well with the drastic effects shown in Figure 1D. The criticism was less of the quality of the images but rather of the fact that they do not seem very representative of the data in Figure 1D.

Lines 80 – 91 still read as if parthanatos is the regulated cell death that is later referred to in the remainder of the manuscript. Again, I would refrain from making the connection to parthanatos. Similarly, the sentence on PARP1 hyper-activation in different diseases on lines 376 – 380 seems to only have a tentative connection with the remainder of the discussion and again seems to hint at parthanatos.

Reviewers' Comments to Author

Reviewer #2 (Remarks to the Author):

The authors have addressed all my concerns with further experimental data and by adding the informations required in the materials and methods section.

Response: Thank you for the positive comment on our manuscript.

Reviewer #3 (Remarks to the Author):

The authors have made serious efforts to respond to all points of criticism and the data and manuscript are much improved. Remaining comments:

The representative images in Figure 1C still do not seem to align well with the drastic effects shown in Figure 1D. The criticism was less of the quality of the images but rather of the fact that they do not seem very representative of the data in Figure 1D.

Response: We appreciate the reviewer's valuable and thoughtful suggestions. The potential drastic effects shown in Figure 1C could be due to the weak fluorescent signal generated by part rather than the complete degradation of the substrate CCF2 by YfiD/EseG-TEM1. To improve the quality of Figure 1C, we increased the image brightness and adjusted the contrast. The merged and enlarged images are also provided to make them intuitive.

Lines 80 – 91 still read as if parthanatos is the regulated cell death that is later referred to in the remainder of the manuscript. Again, I would refrain from making the connection to parthanatos. Similarly, the sentence on PARP1 hyper-activation in different diseases on lines 376 – 380 seems to only have a tentative connection with the remainder of the discussion and again seems to hint at parthanatos.

Response: Thank you for this valuable comment. We apologize that the previous claim on parthanatos needed to be more careful. We have now removed the link to parthanatos in Lines 81-83 and 376.